# Machine learning prediction and tau-based screening identifies potential Alzheimer's disease genes relevant to immunity

Jessica Binder [1], Oleg Ursu [1,7], Cristian Bologa [1], Shanya Jiang [2], Nicole Maphis [2], Somayeh Dadras[2], Devon Chisholm[2], Jason Weick [3], Orrin Myers[1], Praveen Kumar [1], Jeremy J. Yang [1], Kiran Bhaskar [2,4]✉ & Tudor I. Oprea [1,5,6,8]✉

With increased research funding for Alzheimer's disease (AD) and related disorders across the globe, large amounts of data are being generated. Several studies employed machine learning methods to understand the ever-growing omics data to enhance early diagnosis, map complex disease networks, or uncover potential drug targets. We describe results based on a Target Central Resource Database protein knowledge graph and evidence paths transformed into vectors by metapath matching. We extracted features between specific genes and diseases, then trained and optimized our model using XGBoost, termed MPxgb(AD). To determine our MPxgb(AD) prediction performance, we examined the top twenty predicted genes through an experimental screening pipeline. Our analysis identified potential AD risk genes: *FRRS1, CTRAM, SCGB3A1, FAM92B/CIBAR2*, and *TMEFF2*. *FRRS1 and FAM92B* are considered dark genes, while *CTRAM, SCGB3A1*, and *TMEFF2* are connected to TREM2-TYROBP, IL-1β-TNFα, and MTOR-APP AD-risk nodes, suggesting relevance to the pathogenesis of AD.

[1] Department of Internal Medicine, University of New Mexico School of Medicine, Albuquerque, NM 87131, USA. [2] Department of Molecular Genetics and Microbiology, University of New Mexico School of Medicine, Albuquerque, NM 87131, USA. [3] Department of Neuroscience, University of New Mexico School of Medicine, Albuquerque, NM 87131, USA. [4] Department of Neurology, University of New Mexico School of Medicine, Albuquerque, NM 87131, USA. [5] Department of Rheumatology and Inflammation Research, Institute of Medicine, Sahlgrenska Academy at Gothenburg University, 40530 Gothenburg, Sweden. [6] Novo Nordisk Foundation Center for Protein Research, Faculty of Health and Medical Sciences, University of Copenhagen, 2200 Copenhagen, Denmark. [7] Present address: Computational and Structural Chemistry, Merck & Co., Inc., 2000 Galloping Hill Road, Kenilworth, NJ 07033, USA. [8] Present address: Roivant Discovery Sciences, Inc., 451 D Street, Boston, MA 02210, USA. ✉email: kbhaskar@salud.unm.edu; toprea@salud.unm.edu

Except, potentially aducanumab[1], most clinical trials against Alzheimer's disease (AD) focused on the amyloid hypothesis have failed[2–5]. There is a dire need to understand AD's molecular signatures and identify novel therapeutic targets. Genetic analysis of familial AD suggests that point mutations in specific risk genes (e.g., APP, PSEN1, or PSEN2) are known to cause early-onset AD[6]. In the case of more common sporadic AD forms, specific polymorphisms in some genes (e.g., APOE-ε4 or TREM2) are highly correlated to late-onset AD[7–9]. However, changes in these genes do not account for a definitive cause of AD/ADRD. For example, many APOE-ε4 carriers remain disease-free, and some APOE-ε2 carriers develop late-onset AD[10]. Genetic modifiers that override the effects of APOE alleles may explain these paradoxical cases and suggest a need to look further into the AD genetic network.

Moreover, most of the approved and indicated drugs for AD are acetylcholine and glutamate receptor modulators. These drugs, however, only offer symptomatic relief, e.g., temporary improvement of cognitive function and memory loss, and are not disease-modifying medicines[11]. Anti-amyloid antibodies such as solanezumab[12] failed to slow cognitive decline in patients with inherited (autosomal dominant) AD[13]. The accelerated FDA approval for Aducanumab (marketed as Aduhelm®) as AD-modifying treatment in June 2021 remains controversial[14,15], as Biogen must demonstrate clinical benefit in a post-approval trial. As of this writing, gantenerumab has received breakthrough therapy designation by the US Food and Drug Administration, something for which donanemab and solanezumab are also being considered. However, no unequivocal preventatives or cures for AD are currently available, despite the number of approved drugs, ongoing drug applications, and clinical trials.

Many unknowns exist when studying a detrimental heterogeneous disease with multiple categories, even with large amounts of data accumulated from preclinical and clinical studies. Using computational power may be advantageous towards mapping these entangled networks of molecular pathways/genes, finding new targets for therapy, or predicting disease onset, diagnosis, or prognosis at a much faster pace with logical accuracy[16–20]. There are several AD-related machine learning (ML) models previously reported[21]. Transcriptomics was combined with interactomics of RNA-binding proteins to decipher neurodegenerative disorders[22]. Another method trained a deep learning classifier model to recognize and quantify tau burden in the neuropathological assessment of neurofibrillary tangles (NFTs) in post-mortem human brain tissue[23]. There are also a few ML methods that tried to establish gene-disease associations[24–28]. Another study identified the whole-genome spectrum of AD by implementing a Support Vector Machines[29] (SVM) model, classifying collected AD-associated genes in the context of brain-specific functional networks using Genome-Scale Integrated Analysis of Networks in Tissues (GIANT) interface[30]. The identification of differentially expressed genes common in both blood and brain samples from mild cognitive impairment (MCI) and AD patients compared to healthy controls was classified using the LASSO[31,32] (least absolute shrinkage and selection operator) method. A novel AD prediction model based on deep neural networks integrated two heterogeneous datasets: gene expression and DNA methylation profiles[33]. An interpretable ML model for AD diagnosis named sparse high-order interaction model with rejection option (SHIMR) used a weighted sum of short rules[34]. This model also incorporated a rejection function so that physicians could seek other diagnosis methods that are more accurate but may be more costly or invasive when SHIMR is not confident enough to make a diagnosis. While there has been increased interest in ML utilization for AD research, either for novel biomarker/drug target discovery or developing a robust and efficient diagnostic

pipeline, the field is still in its infancy and needs further iterations. Furthermore, to our knowledge, there is no ML algorithm that can mine vast datasets related to AD/ADRD in the public domain and perform meta-analytical prediction of novel AD risk genes.

Here, we report the development of an AD-focused ML model to identify potential AD-associated genes. Starting with data aggregated in the Target Central Resource Database (TCRD)[35,36], we collated 13 distinct datasets, totaling over 261 million attributes (covering several knowledge areas) shown in Supplementary Table 1. The resulting database was integrated into a protein knowledge graph (PKG) using the metapath approach, containing multiple types of entities (nodes) and relationships (edges)[37]. We transform our PKG evidence paths into vectors by metapath matching and convert these data into matrices via gene to disease-association ML features. Metapath-based methods preserve the network structure and gain flexibility in a diverse set of descriptors. To conclude the MPxgb(AD) model, we incorporated the XGBoost[38] algorithm, an optimized gradient boosting tree method, to train and test the model, as well as generate AD-associated gene predictions. Lastly, to determine our MPxgb(AD) prediction performance, we used three different biological screening models to examine the role of the top 20 MPxgb(AD)-predicted genes and their potential correlation to AD-related pathology. Our model identified several potential AD-risk genes: FRRS1, CTRAM, SCGB3A1, FAM92B/CIBAR2, and TMEFF21.

## Results

**Developing the (ProteinGraphML) knowledge graph for metapath-based ML**. TCRD integrates heterogeneous datasets from Illuminating the Druggable Genome (IDG)[39] with an overarching goal to illuminate understudied (dark) protein-coding genes as potential drug targets. TCRD comprises diverse, heterogeneous knowledge about genes, proteins, and small molecules, collated and standardized from various distinct resources[35]. TCRD archives information about protein/gene functions, including text-mined associations from biomedical and patent literature, protein/gene expression data, disease and phenotype associations, compound bioactivity data, and drug-target interactions. A unique aim of the IDG is to assist with the development/druggability of drug targets, assessed via Target Development Level (TDLs)[39]. TDLs are categorized into four development/druggability levels: **Tclin**, **Tchem**, **Tbio**, and **Tdark**. **Tclin** proteins are associated with the known mechanism of action of approved drugs. **Tchem** targets have activities in ChEMBL[40], Guide to Pharmacology[41], or DrugCentral[42] that satisfy the activity thresholds, but no approved drugs. **Tbio** targets have no known drug or small molecule activities that meet the activity thresholds and satisfy one or more of the following criteria: target is above the cutoff criteria for Tdark, the target is annotated with a Gene Ontology Molecular Function or Biological Process[43] leaf term(s) with an Experimental Evidence code. **Tdark** targets have limited information or knowledge about them and include ~30% of the human proteins manually curated at the primary sequence level in UniProt that do not meet the **Tclin**, **Tchem**, or **Tbio** criteria.

While homogeneous graph analytics (e.g., node2vec) yielded advances, heterogeneous graph analytics using meta paths are becoming a powerful approach for modeling complex biological systems. Biological system networks are heterogeneous with multiple nodes and edge types. Developments in heterogeneous[44,45] and biological system networks[37] relationship predictions introduced and formalized a new framework that considers heterogeneity by defining type-specific node-edge paths or meta paths[20]. Although TCRD is maintained primarily as a relational Structured Query Language database, for our ML approach, a subset of TCRD is transformed into a heterogeneous protein-coding gene knowledge

graph (a.k.a. network). By systematically assembling these data, we included data from major areas specific for human protein-coding genes (Fig. 1a): phenotype and disease, pathways, and interactions. Each area has appropriate levels of data, e.g., expression, association, membership, treatments, localization, and gene signatures. A meta path represents a type-specific path pattern between a source node and a destination node. Each instance of a metapath represents a specific chain of evidence of associations between a source and a destination node[20].

For example, Fig. 1a, the metapath {Target — (member of) → PPI (PPI network) ← (member of) — Protein — (associated with) → Disease} summarizes multiple meta paths for PPI data. The PPI data are aggregated from the STRING database (STRINGDb)[46], and the corresponding SQL is: `SELECT protein1_id, protein2_id, score AS combined_score FROM ppi WHERE ppitype = 'STRINGDB'`. Such an example would be {IL1B — (member of) → KEGG Alzheimer's Disease pathway ← (member of) — MAPT — (associated with) → MAPK signaling pathway}. Type-specific metapath counts can be combined using Degree Weighted Path Counts (DWPCs, Eq. 1)

$$DWPC = \sum_{path \in Paths} \left( \prod_{d \in D_{path}} d^{-W} \right)$$

to dampen the effect of highly connected nodes. The original biological system network application used logistic regression and ridge logistic regression. However, we adapted the metapath framework to the extreme gradient boosting (XGBoost)[38]. Contributions from each unique typed meta-path were summarized using *DWPCs* from the input training set for ML. The Methods section provides details regarding ProteinGraphML.

**Focus on Alzheimer's disease**. Exclusively for AD, our Protein-GraphML—termed MPxgb(AD), was trained with AD-focused genes using a training set consisting of 53 AD-associated genes (positives) from the Rat Genome Database and 3,952 genes *not* associated with AD from OMIM (negatives). The Rat Genome Database data, extracted in February 2018, are no longer available. New data, following a similar format, are now accessible[47] from rgd.mcw.edu. The external *test* set contained 23 positively associated AD genes text-mined from DISEASES[48] and 200 random negative genes from DISEASES *not* associated with AD, respectively. The 53 positive genes from the training set and the 20 positive genes from the test set are listed in Supplementary Table 2. Thirteen distinct datasets totaling over 261 million attributes (summarized in Supplementary Table 1) served as input for the initial MPxgb(AD) model. Given the highly imbalanced nature of the training set (more negatives than positives), we addressed this issue in two ways: (1) assign higher weights to the AD-associated (positive label) genes using the "scale_pos_weight" parameter of XGBoost; or 2) generate a balanced training set by sampling with replacement of positives to match the number of negatives. In order to select the best performing model, we used five-fold cross validation (CV) and test set performance to evaluate the *weighted* method—AUC-ROC (area under the curve/receiver operating characteristic) = 0.91/ 0.93 (five-fold CV/test set); and the *balanced* method—AUC-ROC = 0.98/0.62 (five-fold CV/test set). Given better model performance, we selected the *weighted* approach instead of the *balanced* method for the final ML model (Fig. 1c, d). We used cutoffs derived from ROC curve analyses for both models to determine the *sensitivity* for both weighted (0.87) and balanced (0.7) models, and *specificity* for the weighted (0.8) and balanced (0.53) models, respectively. See Supplementary Table 3 for confusion matrices.

The MPxgb(AD) classifier model VIP (variable importance plot) features are as follows: *interactions with proteins mediating inflammatory processes* (JAK2, IL10, and IL2), *response to oxidative stress* (GSTP1), *nervous system development* (BDNF), and *glycolysis* (GAPDH). The drug-induced gene expression perturbation signatures from LINCS[49] were the largest category of features for the weighted AD classifier model. Brain cortex GTEx[50] gene expression and one Reactome[51] pathway (AU-rich mRNA elements binding proteins) were also deemed significant. The list of top 20 features is in Table 1. We then examined contributions of the MPxgb(AD) model features to individual genes within a similarity network. For example, the top predicted AD-associated gene, AKNA (AT-hook-containing transcription factor) displays most of the contributing features of the weighted model, but in a slightly different order (Fig. 1e). Of the 22,549 input features, only 692 contributed to boosted tree building. Of those 692, only 242 features had gain > 0.0001, and the remaining features had gain < 9.99e−05.

Having not archived the *seed number* (a random starting point for ML models that influences results) for the 2018 MPxgb(AD) model, we can no longer reproduce this model. It is however essential to re-assess versioning of datasets and databases with the same input/training sets over time to determine reproducibility and feature importance metrics between various models. Upon running a 10-fold CV for XGBoost on the 2021 version of the database (Supplementary Fig. 1a), AUC-ROC values ranged from 0.8120–0.9889 (mean: 0.8822). XGBoost used between 209 and 249 features out of 22,549 to learn the models. As model training selected different sets of records to train the model in each fold of the 10-fold CV, the list of important features differed between models. The common features among ten models varied from 39 to 61. We used weighted models in the 10-fold CV, and the mean AUC-ROC is comparable to what we reported for the 2018 five-fold CV weighted models (Fig. 1c). Although there were 39–61 features common in 10 models, their rankings were not the same. Depending on the data selected to train the models in 10-fold CV, the contribution of features also varied. In comparison to XGBoost models, LASSO models have a considerable AUC-ROC variance (Supplementary Fig. 1a). Like XGBoost, LASSO regression automatically selects features to learn a model. It used 8–79 features out of 22,549 to learn the models in the 10-fold CV. The common features among ten models varied from 6 to 34.

We then compared model performance by excluding LINCS features from our learning set. After dropping LINCS, the training and test datasets had 3,602 features. Without LINCS, XGBoost models ranged from 0.6023–0.9994 (mean: 0.8701). Comparing the performance of models with/without LINCS (Supplementary Fig. 1a), the lower bound of the AUC dropped, but the mean AUC and upper bound remained similar. XGBoost used 181–206 features out of 3,602 to learn the models. The common features among ten models varied from 56 to 82. After dropping LINCS, XGBoost models had more features in common. There was no significant change in the lower and upper bound of AUC when dropping LINCS in LASSO models, but the mean AUC increased, 0.6807–0.9977 (mean: 0.9085). LASSO used 34–446 features out of 3,602 to learn the models. The common features among ten models varied from 18 to 145. After dropping LINCS, LASSO comparatively used more features to learn models, and the models had more features in common. Among the top 20 features of LASSO and XGBoost models without LINCS, 2–4 features were the same. We did not observe significant changes in the count of common features among XGBoost and LASSO models after dropping LINCS. When comparing 2018 prediction lists with 2021 results, there was a decent overlap when examining the top 100 predicted genes among various models (Supplementary Fig. 1b), thus providing

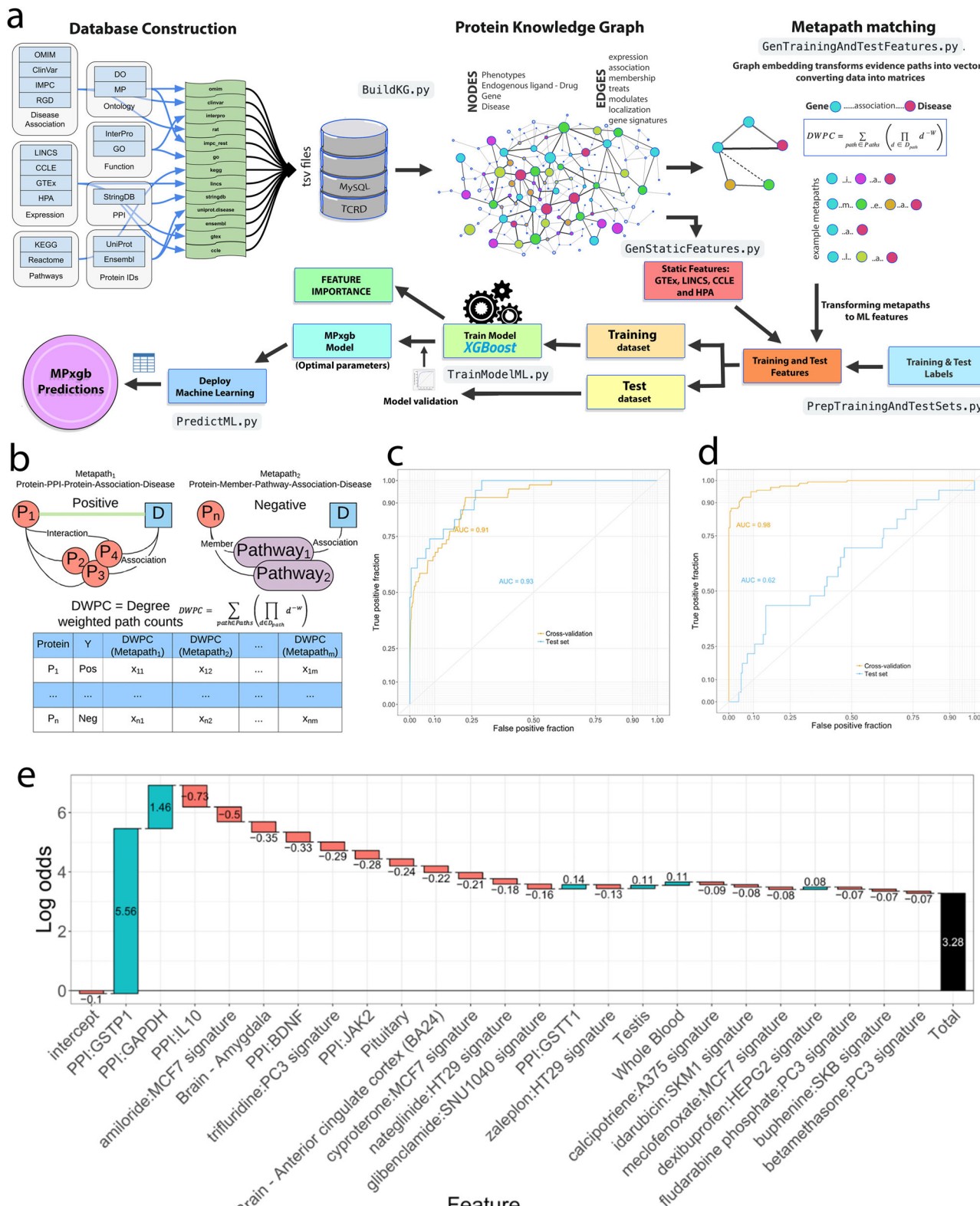

**Fig. 1 TCRD knowledge graphs concept and overview of the meta-path-XGBoost algorithm, MPxgb(AD), and workflow.** Centered around the knowledge tree, this concept was essential in selecting data types (Table 1) for the ML algorithms used to impute AD associations for potential proteins/genes. **a**. Transformation of knowledge graph to ML-ready dataset and training of the model. An example metapath: {Target — (member of) → PPI (protein–protein interaction network) ← (member of) — Protein — (associated with) → Disease} summarizes multiple metapaths for PPI data. **b**. Evidence weighting by degree-weighted path count (DWPC). **c**, **d**. Five-fold cross validation and test set performance are used to evaluate a weighted method (left) AUC-ROC = 0.91/0.93 (five-fold CV/test set) and balanced method (right) AUC-ROC = 0.98/0.62 (five-fold CV/test set) to select the best performing model. **e**. Feature importance prediction for the AKNA-AD association.

**Table 1 Top 20 features used in the boosted trees of the AD-focused MPXgb model.**

| Feature | Data source | Gain |
|---|---|---|
| PPI:GAPDH | STRING | 0.27852 |
| PPI:IL10 | STRING | 0.08283 |
| PPI:GSTP1 | STRING | 0.03823 |
| Darunavir:HELA signature | LINCS | 0.03732 |
| Aminosalicylic acid:MCF7 signature | LINCS | 0.03413 |
| Blonanserin:PC3 signature | LINCS | 0.03004 |
| Amiloride:MC7 signature | LINCS | 0.02594 |
| Trifluridine:PC3 signature | LINCS | 0.02048 |
| Brain - Anterior cingulate cortex (BA24) | GTEx | 0.01866 |
| Travoprost:HA1E signature | LINCS | 0.01848 |
| Regulation of mRNA stability by proteins that bind AU-rich elements | Reactome | 0.01775 |
| PPI:JAK2 | STRING | 0.01729 |
| PPI:BDNF | STRING | 0.01684 |
| PPI:IL2 | STRING | 0.01593 |
| Levosulpiride:PC3 signature | LINCS | 0.01556 |
| Azelaic acid:MC7 signature | LINCS | 0.01538 |
| Saquinavir:A549 signature | LINCS | 0.01534 |
| Cyproterone:MC7 signature | LINCS | 0.01411 |
| Vorinostat:HT115 signature | LINCS | 0.01138 |
| Trifluoperazine:WSUDLCL2 signature | LINCS | 0.01092 |

The Gain highlights the relative contribution of each feature to the model.

**Table 2 A list of the top 20 genes predicted from the AD-specific MPXgb model.**

| UniProt ID | HGCN symbol | Predicted probability |
|---|---|---|
| Q7Z591 | AKNA | 0.95666 |
| Q6ZNA5 | FRRS1 | 0.65087 |
| Q8WXH6 | RAB40A | 0.58811 |
| Q14849 | STARD3 | 0.53379 |
| Q8WXW3 | PIBF1 | 0.52237 |
| O95881 | TXNDC12 | 0.47394 |
| Q96QR1 | SCGB3A1 | 0.45196 |
| Q9UN36 | NDRG2 | 0.43148 |
| Q8ND76 | CCNY | 0.42928 |
| Q6ZTR7 | FAM92B | 0.42337 |
| Q14957 | GRIN2C | 0.38404 |
| Q8IVH2 | FOXP4 | 0.34215 |
| O43791 | SPOP | 0.34097 |
| Q6ZVW7 | IL17REL | 0.32126 |
| O95727 | CRTAM | 0.30046 |
| Q9UIK5 | TMEFF2 | 0.30017 |
| Q9BYV7 | BCO2 | 0.29885 |
| P61968 | LMO4 | 0.29783 |
| Q8N6C8 | LILRA3 | 0.28329 |
| Q8IWA5 | SLC44A2 | 0.276923 |

The Predicted probability column is the XGboost classifier probability that a particular gene belongs to the "AD positive" class.

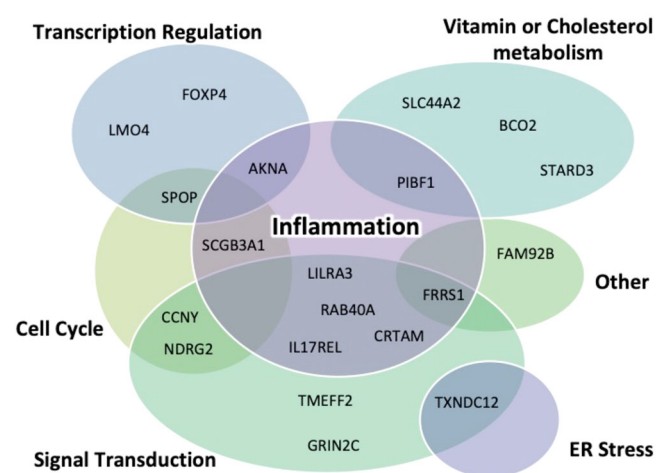

**Fig. 2 Venn diagram of the top potential twenty genes predicted.** Each gene target was designated to each category after in-depth manual literature review (see Supplementary Table 2). The majority of the predicted MPxgb(AD) genes appear to correlate mostly with innate immunity and signal transduction pathways. Transcription regulation, vitamin or cholesterol metabolism, cell cycle, and ER stress are other pathways the predicted MPxgb(AD) genes show functional associations.

41 true negatives as positive (i.e., false positives). Since only one in three *de novo* predicted genes is likely to be a true positive, we anticipated up to 7 of the top 20 genes would be experimentally confirmed. To further test the validity of the ML model, we also examined the bottom 10 MPxgb(AD) predicted genes.

**Developing the biological screening system for AD-specific predictions.** To understand more about the top twenty MPxgb(AD) predicted genes, we manually searched the literature and clustered them into seven major functional categories: inflammation, signal transduction, transcription regulation, vitamin or cholesterol metabolism, cell cycle, ER stress, and other pathways (Fig. 2). From previous indications of these MPxgb(AD) predicted genes, we concluded using three logically defined AD model systems to screen potential implications in AD-related pathology (Supplementary Fig. 2). Utilization of AD patient-derived induced pluripotent stem cells (iPSCs)[52] has also shown to be useful disease relevance models[53]. First, using a well-defined differentiation protocol[54], we assessed the mRNA levels of each top 20 MPxgb(AD) predicted gene in human iPSC derived neurons (iPSNs) derived from sporadic AD (sAD2.1; 83 years old male) and compared them with control iPSNs (AX0018—from Axol Bioscience; #ax0018-kit; 74 years old male). Ideally, patient samples are a precise resource for experimental phenotypic analysis. Therefore the second screening model analyzes mRNA levels for each top 20 MPxgb(AD) predicted gene isolated from clinically diagnosed sporadic AD and healthy control post-mortem brain samples (temporal cortices). Consecutively, from the same post-mortem brain samples, protein levels are also analyzed between the two groups.

An increasing number of studies suggest that tau pathology correlates more with cognitive decline[55–57] and brain atrophy[58] than amyloid pathology. Microtubules are dynamic filamentous structures that comprise a large portion of the cellular cytoskeleton[55]. Microtubules' primary functions are to determine the cellular shape and transport molecular cargo inside cells. Microtubule-associated protein tau (MAPT) is predominantly found in neuronal cells and responsible for stabilizing microtubule structure and facilitating axonal transport[59]. However, tau also undergoes many post-translational modifications, with

significant reproducibility confidence. Furthermore, when including LINCS, XGBoost performs slightly better, whereas LASSO performs slightly better without LINCS. Based on this retrospective evaluation (2021 vs 2018 models), and for the sake of interpretability, we focus on results from XGBoost as the algorithm of choice for the ML model.

Albeit the 2018 MPxgb(AD) model provided probabilities of AD-association for thousands of genes, we focused on the top 20 genes, listed in Table 2. While correctly predicting 20 out of 23 positive genes (true positives), the *weighted* model also predicted

phosphorylation being one of the well-studied modifications. Excessive phosphorylation of tau reduces its binding to microtubules, causing disassembly of microtubules[60]. The accumulation of hyperphosphorylated tau as paired-helical filaments (PHFs) or straight filaments (SFs), eventually leading to NFTs, represents a prominent pathological hallmark of many tauopathies, including AD[55,60]. The National Institute on Aging and Alzheimer's Association (NIA-AA) currently uses NFTs as a diagnostic criterion[61]. Among different phosphorylation sites, hyperphosphorylation at $Ser^{199}/Ser^{202}/Thr^{205}$ site specifically recognized[62] by the antibody AT8 termed $pS^{199}/pS^{202}/pT^{205}$; and at $Thr^{231}$ site, specifically recognized[60] by the antibody AT180 termed $pT^{231}$, are known to be the earliest epitopes during disease progression[63]. We previously reported that conditioned media derived from activated microglia induces tau phosphorylation in neurons via activation of p38 Mitogen-Activated Protein Kinase (p38 MAPK)[64–66]. Given that most of our top 20 MPxgb(AD)-predicted genes are involved in inflammation (Table 2; Supplementary Note 1), it is logical to determine any potential role in inflammation-induced tau pathology. Therefore, the third screening model includes investigating inflammation-induced tau phosphorylation following siRNA-mediated individual knockdown of each of the top 20 MPxgb(AD) predicted genes within SH-SY5Y human neuroblastoma cells. In this model, first, each of the top 20 MPxgb(AD) predicted genes are knocked down by siRNA, followed by treatment of conditioned media (CM) derived from LPS (100 ng/ml)-primed microglia. We then examined the levels of phospho(p)-$Ser^{199}/pSer^{202}$ (AT8 site) and $pThr^{231}$ (AT180 site) in undifferentiated SH-SY5Y cells.

**mRNA levels for four of the top 20 MPxgb(AD) predicted genes are altered in human sAD2.1 iPSNs compared to control AX0018 iPSNs.** We first quantified mRNA levels within AD-specific neuronal cell types. As a first step in confirming the AD-related tau phenotype and neuron differentiation of sAD2.1 iPSNs, we performed double immunofluorescence analysis for AT8 or AT180 tau and βIII-tubulin positive cells within sAD2.1 and AX0018 iPSNs (Fig. 3a, b). As previously reported[52], we validated elevated levels of AT8 and AT180 by performing Western blot analysis. We observed a significant increase in AT8/Beta-Actin and AT180/Beta-Actin ratios in sAD2.1 iPSNs compared to control AX0018 iPSNs (Fig. 3c, d). These results confirmed the purity of sAD2.1, which showed expected levels of hyperphosphorylated tau. Next, we performed qRT-PCR to determine the mRNA levels of all twenty MPxgb(AD) predicted genes. Three of the twenty genes (RAB40A, SCGB3A1, and TMEFF2) showed significant upregulation of mRNA levels in sAD2.1 iPSNs compared to control AX0018 iPSNs (Fig. 3e). Only one gene, FRRS1, showed significantly reduced mRNA levels (Fig. 3e). Together, these results provide information on gene expression status within patient-derived neurons and four genes that have not previously been known to be involved in AD/ADRD pathogenesis.

**In human post-mortem AD brains, five of the top 20 MPxgb(AD) predicted genes are altered at the mRNA levels, and nine are at the protein level.** To quantify the top 20 MPxgb(AD) predicted genes levels on a broader scale, we examined whether or not mRNA and proteins of these predicted genes are altered in the human AD brain tissue (Supplementary Table 4). Post-mortem cortical samples include other non-neuronal cells besides neurons. Therefore, providing more insight towards MPxgb(AD) predicted genes and potential relevance to AD pathogenesis at the organ level. Here we show five (AKNA, FRRS1, NDRG2, FAM92B, and SLC44A2) mRNA levels of the MPxgb(AD) predicted genes (except for FRRS1) are significantly downregulated in human AD than

healthy controls. mRNA levels of the remaining fifteen genes were unaltered (Supplementary Fig. 3). On the other hand, protein levels for nine of the MPxgb(AD) predicted genes, FRRS1, STARD3, PIBF1, TXNDC12, FAM92B, FOXP4, SPOP, CTRAM, and LILRA3, were significantly higher in AD brains compared to controls (Fig. 4). Two of them (SCGB3A1 and SLC44A2) were undetectable in AD or control brain samples, possibly due to antibody failure (Fig. 4). These results confirm the sAD2.1 iPSN data (Fig. 3) and suggest that some of the ML-predicted genes regulated at the translational level contribute to AD pathogenesis.

**siRNA-mediated knockdown of CRTAM, FOXP4, GRIN2C, LILRA3, PIBF1, SCGB3A1, and TXNDC12 reduce inflammation-induced tau phosphorylation.** Since most of the antibodies for novel AD targets were relatively new, i.e., not sufficiently tested or optimized, we first validated commercially available antibodies for their target recognition via immunofluorescence and Western blot analysis. We note that not all antibodies worked well for validation via Western blots or immunofluorescence staining (Supplementary Fig. 4). Yet, we observed 13 out of 20 targets showing significant siRNA-mediated knockdown; 3 out of 20 showing moderate knockdown, and 4 out of 20 showed no knockdown via Western blot analysis (Supplementary Fig. 4). We could see a substantial reduction in target immunoreactivity for those showing moderate to no knockdown via Western blot (e.g., BCO2, FAM92B, RAB40A, and SCGB3A1, etc.), immunofluorescence analysis (Supplementary Fig. 4). Overall, for most of the target genes, siRNA-mediated knockdown shows a substantial reduction in each target gene either via western-blot or immunofluorescence analysis (Supplementary Fig. 4).

Next, we used an in vitro neuroinflammation AD model to characterize potential pathological tau mechanistic associations for the predicted genes. The model uses undifferentiated SH-SY5Y cells and treatment of condition media (CM) derived from BV2 cells (murine neonatal microglia that were raf/myc-immortalized)[67]. This approach was similar to our previously published cell culture model where we treated human tau expressing mouse neuroblastoma (N2a) cells with CM from LPS-primed RAW 264.1 macrophages[68]. When SH-SY5Y's are treated with CM, this induces tau phosphorylation at AT8 and/or AT180 sites[64,65] (Fig. 5). Briefly, on the same day, we knocked down each of the 20 genes, individually with the respective siRNA targeting gene of interest (GOI), and separately prime BV2 cells with lipopolysaccharide-S (LPS, an endotoxin) for 24 h. The next day, we replaced 50% of the SH-SY5Y cell culture media with the same volume of BV2 microglial CM (see Supplementary Fig. 2). We then assessed $pS^{199}/pS^{202}$ (AT8) and/or $pT^{231}$ (AT180) tau levels (Fig. 5). When a reduction in AT8+ and AT180+ is observed, this indicates that the target gene contributes to AD-related tau pathology. On the contrary, if AT8+ and AT180+ levels increase, this suggests that target genes are protective against tau pathology. Strikingly, siRNA-mediated knockdown of PIBF1, TXNDC12, SCGB3A1, NDRG2, GRIN2C, FOXP4, CRTAM, LMO4, and LILRA3 significantly reduced the levels of inflammation-induced AT8+ and AT80+ tau phosphorylation (Fig. 5). While 9 out of 20 knockdown target genes significantly reduced phosphorylated tau, we note some caveats within siRNA knockdown methods. Potential off-target hits, relatively low target expression in proliferating SH-SY5Y cells/in CNS, incomplete knockdown, or being a regulator of amyloid-β rather than tau may explain why not all MPxgb(AD) predicted genes showed an effect in reducing/increasing inflammation-induced tau phosphorylation (e.g., BCO2, Supplementary Fig. 4). Nonetheless, nine of the predicted genes did affect AT8/AT180 positive tau and are likely to be risk genes in inducing AD-related hyperphosphorylation of tau.

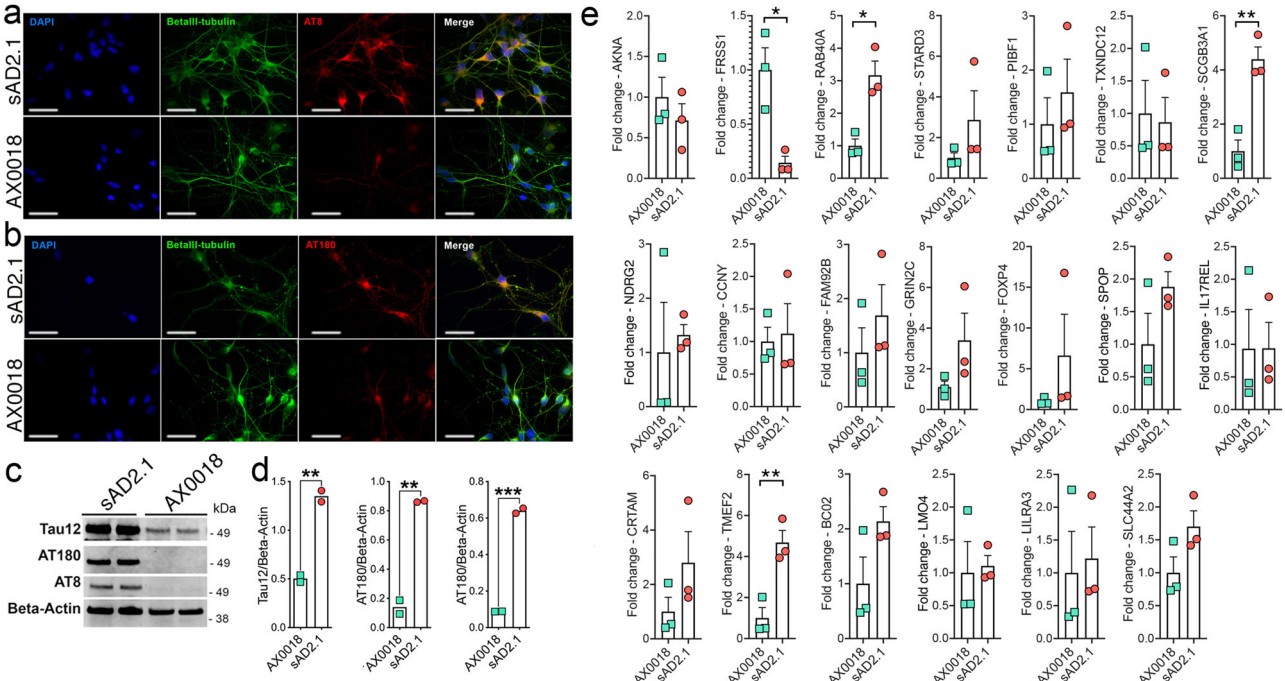

**Fig. 3 mRNA for four of predicted genes are altered in human sAD2.1 iPSNs compared to control AX0018 iPSNs. a, b.** Quantitative immunocytochemistry showing significant increase of pS199/pS202 (AT8) tau in βIII-tubulin (neuronal marker) positive sAD2.1 iPSNs compared to control AX0018 iPSNs. Scale bar: 20 μm. **c, d**. Western blot and quantification showing significantly increased levels of total, pS199/pS202 (AT8), and pT231 (AT180) positive tau levels in sAD2.1 iPSNs compared to control AX0018 iPSNs. (Raw blots are shown in Supplementary Fig. 9). **e**. Quantitative real-time PCR analysis showing two folds increase in the mRNA levels of RAB40A $p$ value = 0.0232, SCGB3A1 = 0.0053, TMEFF2 = 0.0096, and a fold decrease in FRRS1 = 0.0431 in sAD2.1 iPSNs compared to AX0018 control iPSNs. Data shown are mean ± s.e.m; Two-tailed $t$ tests welch-corrected; *$p < 0.05$; **$p < 0.01$, ***$p < 0.005$, $n = 3$ biological replicates; $n = 3$ technical replicates.

**Some of the bottom MPxgb(AD) predicted genes show AD relevance but are not related to immunity/inflammation.** Our MPxgb(AD) model captures cross-modality features from heterogeneous datasets to model a complex system. The ML model trained on both positive and noisy negative data because a true negative set is unavailable. We consider the negative training set "noisy" since it will likely include negative and positive examples. We assume that AD-associated genes can train ML models even with a noisy negative set. We anticipate that genes predicted to have similar biological context from meta paths might also be associated with AD. Therefore, we assigned higher weights to positive examples to force learning from positives more than negative ones. Among the top 20 VIP features selected by the MPxgb(AD) classifier (Table 1), for example, there are protein–protein interactions (PPIs) for inflammatory process mediators that are in the positive training set (*JAK2, IL10,* and *IL2*), as well as PPIs with the oxidative stress response protein (*GSTP1*). These PPIs suggest *infection*, which is when oxidative stress and inflammation co-occur (e.g., phagocytes producing reactive oxygen species).

PPIs with a nervous system development gene (*BDNF*) may be indicative of neurodegeneration. Multiple drug-induced gene expression perturbations from the LINCS dataset were the largest category of features selected by this model. Brain cortex expression (cingulate cortex, area A24) via GTEx and one Reactome pathway (AU-rich mRNA elements binding proteins) were also among the prioritized features. Thus, the weight of these features contributed heavily to similar networks within the positive examples. Our MPxgb(AD) model does not distinguish an AD-association gradient but predicts neurological inflammatory associations instead. Thus, the bottom (lowest probability) predicted genes don't necessarily lack AD association. Instead, we

suspect that the bottom predicted AD-associated genes might constitute a different cluster network within pathways, molecular functions, and pathogenesis.

We then queried the top 20 predicted genes to see if they were clustered into different biological processes compared to the bottom 20 predicted genes by conducting a GO enrichment analysis through two additional resources, PANTHER[69] and EnrichR[70]. Among the top 20 genes, only *GRIN2C*, known to play a role in neurological pathways, was a PANTHER hit. In contrast, the bottom genes were clustered into signaling pathways, oxidative stress response, and DNA replication. (Supplementary Fig. 5a, b). Human Wikipathways results from EnrichR show similar trends in the top 20 genes, with four significantly enriched terms: vitamin A and carotenoid metabolism; hedgehog signaling pathway; NO/cGMP/PKG mediated neuroprotection; and synaptic signaling pathways associated with autism spectrum disorder. The top trending pathways were in phosphodiesterases in neuronal function, Alzheimer's Disease, primary cilium development, and mRNA processing.

On the contrary, the bottom genes had five significant enriched terms; TGF-beta signaling pathway, homologous recombination, globo sphingolipid metabolism, intraflagellar transport proteins binding to dynein, and DNA replication. Overall, the top twenty MPxgb(AD) seem relevant in neurological pathways[71]. In contrast, the bottom twenty are relevant in metabolism and DNA mechanisms; this does not omit neurological processes (Supplementary Fig. 5c–f; see list of bottom ten genes predicted from the MPxgb(AD) in Supplementary Table 5).

To further distinguish the notion of a non-gradient distribution (AD-associated to nonAD-associated) in our AD-association predictions, we ran the bottom ten MPxgb(AD) predicted genes through the first two biological screening methods. First, we

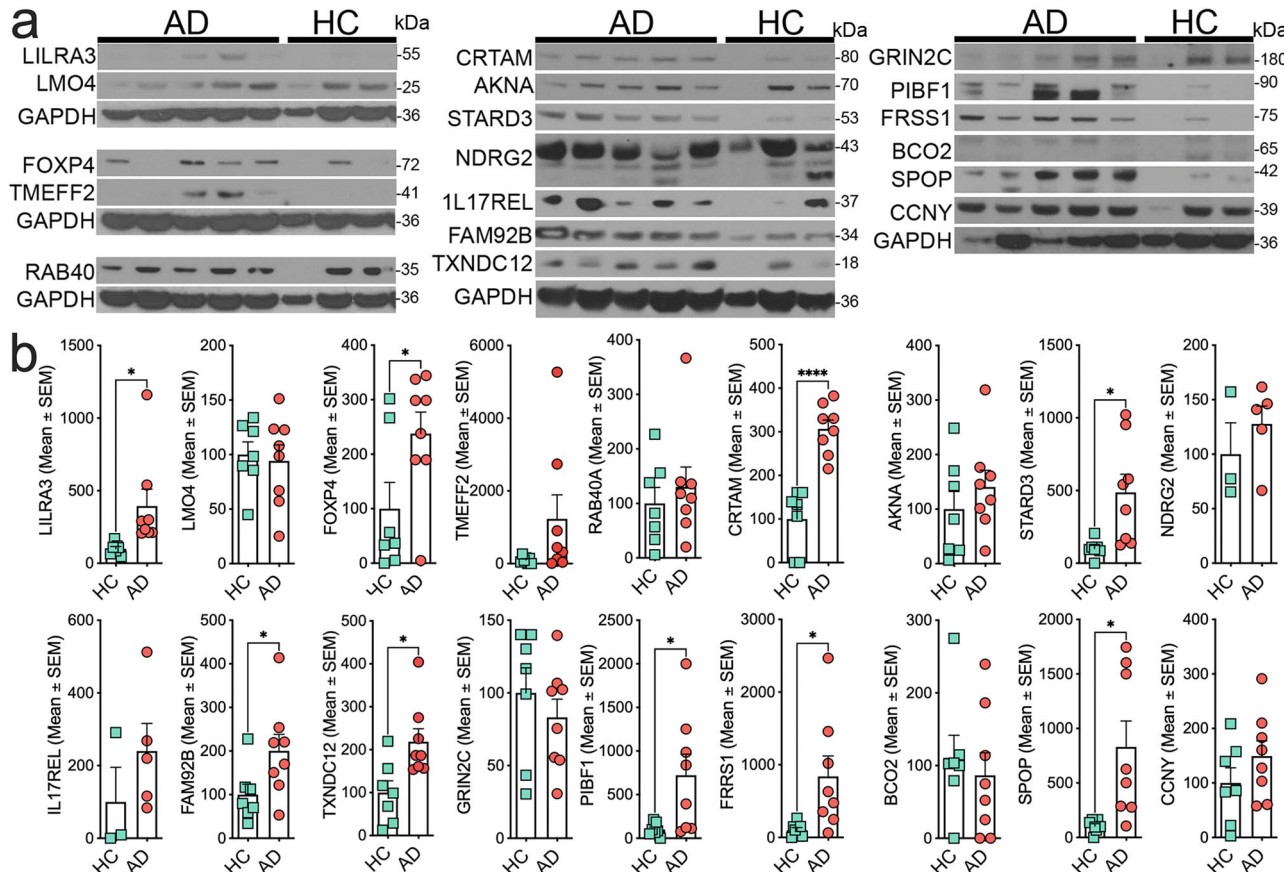

**Fig. 4 Nine proteins relevant to inflammation, transcription regulation, and metabolism are significantly increased in post-mortem autopsy brains of human sporadic Alzheimer's disease. a**, **b**. Western blot and quantifications showing significantly elevated levels of predicted proteins relevant to inflammatory pathways (PIBF1, CRTAM, FRRS1, and LILRA3), transcriptional regulation (FOXP4 and SPOP), metabolism (PIBF1 and STARD3), and others (TXNDC12 and FAM92B) in post-mortem temporal cortical samples of sporadic AD compared to age-matched healthy controls (HC). Note that SCGB3A1 and SLC44A2 were not detectable. Individual gels are separated and aligned with their corresponding loading control (GAPDH). The middle gel was stripped and reprobed accordingly. (Raw blots are shown in Supplementary Fig. 10). Data shown are mean ± s.e.m of ratio to consecutive loading control GAPDH; significance was determined using two-tailed *t* tests Welch-corrected; *p < 0.05; **p < 0.01, (P values for FRRS1 = 0.0340, STARD3 = 0.0176, PIBF1 = 0.0363, TXNDC12 = 0.0122, FAM92B = 0.0456, FOXP4 = 0.0474, SPOP = 0.0183, CRTAM = < 0.0001, and LILRA3 = 0.0392) n = 7 healthy controls and n = 8 sporadic AD).

analyzed the mRNA levels in iPSNs, then measured the mRNA and protein levels in human brain autopsy tissue of the ten least probable AD-associated genes from our MPxgb(AD) model. We determined no significant differences in mRNA levels when comparing sAD2.1 iPSNs to control AX0018 iPSNs (Supplementary Fig. 6). However, *ACSM5* mRNA levels were significantly downregulated in human AD brain tissue versus healthy controls (Supplementary Fig. 7). We also found two significant alterations at the protein level, *STK32B* and *PFKFB2*, comparing human AD brain tissue to healthy controls (Supplementary Fig. 8, raw blots are shown in Supplementary Fig. 12). We further examined these three gene hits by manually curating information derived from Pharos[36], which pulls data from over 60 resources. These three genes, *ACSM5*, *STK32B*, and *PFKFB2*, are not associated with inflammatory processes. Instead, they are involved in fatty acid beta-oxidation, sweet taste signaling, and glycolysis regulation (Supplementary Note 1). It is not clear whether these three genes indirectly influence immune functions. Their changes in AD iPSNs/autopsy samples require further investigation.

Our top predicted MPxgb(AD) genes do not represent absolute AD-associated ranking, nor are the bottom genes least likely to be AD-associated. The difference between top-predicted versus bottom-predicted genes is in their similarity cluster network

(inflammatory vs. metabolic pathways) rather than a definitive verdict regarding AD-association or lack thereof. For complex diseases such as AD/ADRD, more optimization is necessary to build accurate disease-associated gene classification models. Since MPxgb(AD) training is based on similarity networks (meta paths), this model does not produce a ranked list of AD-associated genes. A "ground truth" (confirmed true positives and true negatives) training set would be required to create a binary model. To the best of our knowledge, such gene lists are not available for AD.

**Correlational analysis suggests that two (SCGB3A1 and CRTAM) of the top 20 predicted MPxgb(AD) genes are relevant to immunity.** To finalize and logically correlate all three biological screening models (human iPSNs, human post-mortem samples, and inflammation-induced phosphorylated tau in SH-SY5Y cells), we generated a ranking system to distinguish each experiment's outcome, and the strength of association of each of the top 20 MPxgb(AD) predicted genes. The statistically significant digits are assigned a level unit corresponding to the *p* values and displayed as a heatmap (Fig. 6a). Note, this ranking system does not distinguish between increased or decreased experimental levels. For example, * (*p* value ≤ 0.05) represents one unit level or point;

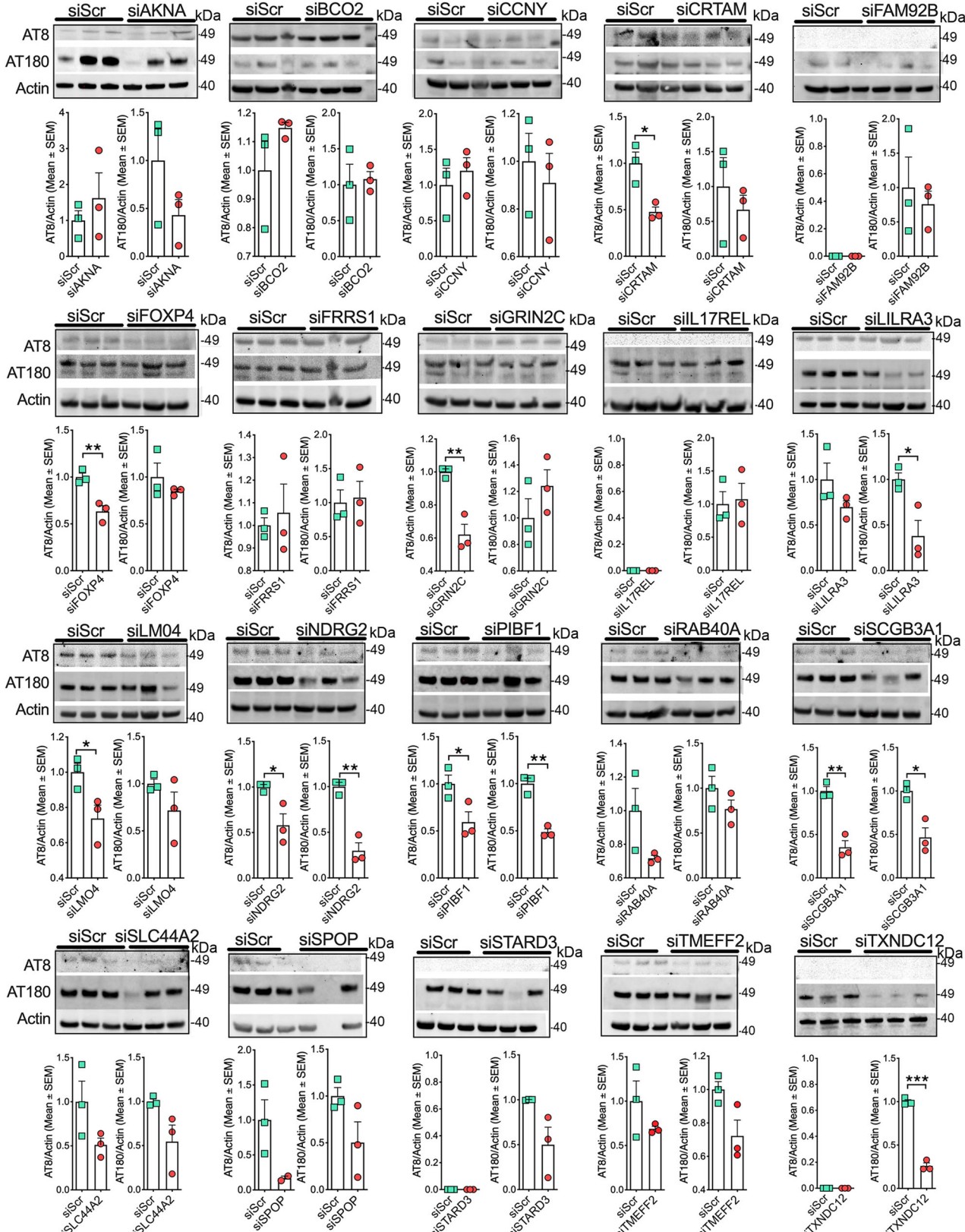

**Fig. 5 siRNA-mediated suppression of CRTAM, FOXP4, GRIN2C, LILRA3, PIBF1, SCGB3A1, and TXNDC12 in SH-SY5Y human neuroblastoma cells reduces inflammation-induced pS199/pS202 (AT8) and/or pT231 (AT180) tau phosphorylation.** SH-SY5Y cells were transiently transfected with either targeted GOI or scrambled siRNAs. After 48 h, cells were treated with CM (at 50%) derived from BV2 microglial cells. After 24 h, the lysates from SH-SY5Y cells were prepared and Western blot was performed for AT8 and AT180 antibodies and actin as loading control. Note that the levels of AT8 and/or AT180 are significantly altered in SH-SY5Y cells upon knockdown of MPxgb(AD) predicted *CRTAM, FOXP4, GRIN2C, LILRA3, PIBF1, SCGB3A1,* and *TXNDC12* genes. (Raw blots are shown in Supplementary Fig. 11). Data shown are mean ± s.e.m; unpaired $t$ tests; *$p < 0.05$; **$p < 0.01$, ***$p < 0.005$, $n = 3$ biological replicates and $n = 3$ technical replicates).

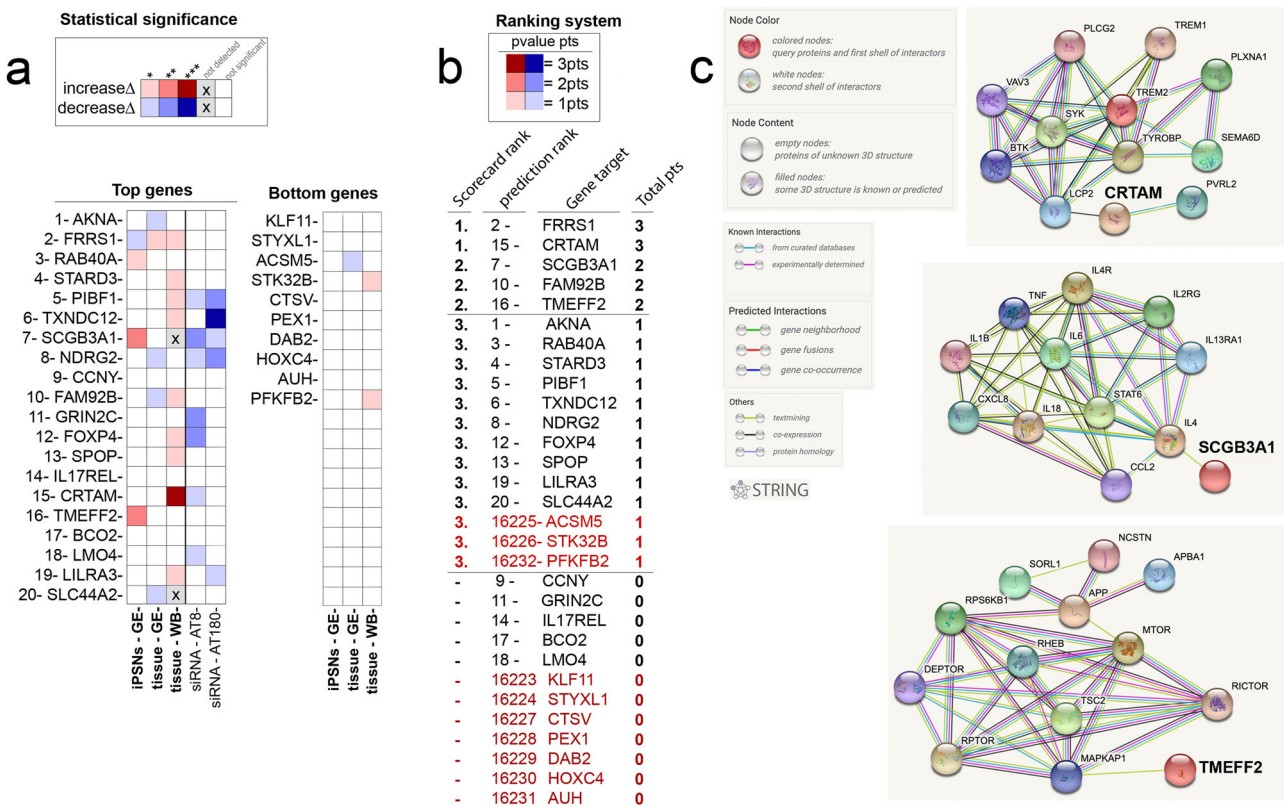

**Fig. 6 Our analysis identified *FRRS1, CTRAM, SCGB3A1, FAM92B/CIBAR2,* and *TMEFF2* as potential AD risk genes. Two (*CRTAM* and *SCGB3A1*) of the validated genes are relevant to immune mechanisms involving TREM2-TYROBP and IL-1β/TNFα axis, respectively. Two (*FRRS1* and *FAM92B*) are considered dark genes. *TMEFF2* is involved in MTOR-APP regulation. a**. Heatmap showing the summary of the top genes vs bottom genes, via three different validations done on all twenty MPxgb(AD) predicted genes, as well as bottom predicted genes. The heat map colors represent the *p* value statistical significance. * = *p* value ≤ 0.05; ** = *p* value ≤ 0.005; *** = *p* value ≤ 0.0005. (1) The first row shows the results from the iPSNs-based validation where the mRNA levels of GOI are either increased (red), decreased (blue) or no change (white) in sAD2.1 iPSNs compared to control AX0018 iPSNs. (2) The second row is mRNA levels for the MPxgb(AD) predicted GOI are either increased (red), decreased (blue) or no change (white) in human autopsy brains. The third row is the results from the human autopsy brains of sporadic AD vs age-matched healthy controls, where the MPxgb(AD) predicted proteins of interest (POI) are either increased (red), decreased (blue), not detectable (gray with an x) or not significant (white). (3) The last two rows show the results from SH-SY5Y siRNA knockdown experiments, where the knockdown of MPxgb(AD) predicted GOI significantly reduced AT8 (blue) and/or AT180 (blue) levels. **b**. * Corresponds with statistical significance levels in (**a**) and represents one unit level or point; ** denote two points, and *** represents three points. Thus, the ranking system is determined by the *p* values and sets each GOI with a total score calculated from each screening experiment. Note, the siRNA experiments were not included in the final rank score. Moreover, the final list is labeled as top ranked with 3pts total score being the highest (also labeled with their original predicted rank number), *FRRS1* and *CTRAM* are tied for our top significant AD-relevant genes. With 2pts total score, *SCGB3A1, FAM92B/CIBAR2,* and *TMEFF2* are ranked second. **c**. STRING protein network connectivity analysis of the three (*CRTAM, SCGB3A1,* and *TMEFF2*) of the top five validated genes show interactions with well-established TREM2-TYROBP, IL-1β/TNFα, (both immune axes) and MTOR-APP relevant to AD, respectively.

**(*p* value ≤ 0.005) denote two points (Fig. 6b). We recall that siRNA can potentially have off-target events.72 Therefore, we excluded the knockdown experiments from the total score.

We totaled the points from iPSNs and tissue analyses, ranking the predicted genes from highest to lowest score (Fig. 6b). Upon performing this correlational analysis, *FRRS1, CRTAM, SCGB3A1, FAM92B,* and *TMEFF2* showed the expected trend. The bottom 10 of the MPxgb(AD) predicted genes had three 1-point gene hits (*ACSM5, STK32B,* and *PFKFB2*), when comparing the level of significance in the iPSNs and tissue analysis. However, we did not consider ≤ 1 point to be significant enough to suggest AD-relevance. Among the top five ranked genes, *FRRS1 and CRTAM* show the highest level of AD-association. *FRSS1*, categorized as Tdark in the IDG classification, is a Ferric-chelate reductase that reduces $Fe^{3+}$ to $Fe^{2+}$ before transport from the endosome to the cytoplasm[72]. Interestingly, age-related dysregulation of brain iron homeostasis leads to abnormal iron accumulation[73]. Another study

suggests that under proinflammatory responses, greater uptake of iron occurs in brain microglia[74]. Perhaps defective FRRS1 might play a role in the bioaccumulation of iron deposits from the brain, which has been suggested in AD etiology[75].

FAM92B, recently renamed CIBAR2 (CBY1 Interacting BAR Domain Containing 2), is also categorized as Tdark; it may facilitate ciliogenesis via cell projection organization[76]. CRTAM, Cytotoxic, and Regulatory T Cell Molecule encodes a type I transmembrane protein with V and C1-like Ig domains[77]. The *CRTAM* gene is upregulated in CD4+ and CD8+ T cells[78]. CRTAM$^{(-/-)}$ mice exhibited reduced protective immunity against viral infection and impaired autoimmune diabetes induction in vivo[79].

Analyzing the neighboring nodes and closely interacting proteins of CRTAM and SCGB3A1 (both, Tbio) in the STRING[80] database, CRTAM and SCGB3A1 are directly/indirectly interacting with the well-established immune axes IL-1β/TNFα and TREM2-TYROBP,

respectively, which are relevant to AD (Fig. 6c). Subsequently, we also note that CRTAM has a third and fourth-degree relationship to the other known AD-associated genes, *ITPR1* and *VPS35*, respectively. VPS35[81] (VPS35 Retromer Complex Component) is related to GPCR signaling and endocytosis pathways. Furthermore, ITPR1[82], Inositol 1,4,5-Trisphosphate Receptor Type 1, is involved in downstream B-Cell receptor signaling events, GPCR signaling, as well as ion (e.g., calcium) channel activities. TMEFF2 (also categorized as Tbio) has been shown to be protective in AD via binding of amyloid-β oligomers[83]. In addition, this study suggests an endurance element for hippocampal and mesencephalic neurons[84]. TMEFF2 is also connected to MAPKAP1[85], a subunit of mTORC2 (Target of rapamycin complex 2) that regulates cell growth and survival in response to hormonal signals[85] indirectly connected to APP (Fig. 6c). In summary, these results suggest proteins relevant to inflammation/cytokine activity, innate immunity, and amyloid-β processing axes are potential AD/ADRD pathogenic pathways, which were identified and screened through an AD-focused MPxgb model.

## Discussion

By combining a large-scale dataset (the TCRD subset shown in Supplementary Table 1) with the graph-based metapath formalism and the ML XGBoost algorithm, MPxgb(AD), we generated an AD-focused model built upon the Rat Genome Database-based list of 53 positive AD-associated genes. In this study, we reported an AD-focused ML model and screened 20 top and 10 bottom genes in a series of three independent biological AD model systems. With ML guidance, we identified potential gene targets previously not associated with AD relevant to immunity. Our qRT-PCR analysis shows significantly altered mRNA levels in the sporadic AD iPSNs in four of the top twenty predicted genes. Five of the top 20 genes are significantly altered at the mRNA level, and nine are upregulated at the protein level in human autopsy AD brains. siRNA-mediated knockdown confirmed nine genes significantly reduced AD-relevant tau hyper-phosphorylation on AT8 and AT180 sites.

Five (*FRRS1*, *CTRAM*, *SCGB3A1*, *FAM92B/CIBAR2,* and *TMEFF2*) of the top 20 genes were ranked the most significant distinctions between AD samples vs. control. All five top-selected genes lack known associations with small molecules and drugs, having been categorized as Tbio or Tdark (FRRS1 and FAM92B/CIBAR2)[86]. Two genes (*CTRAM* and *SCGB3A1)* are associated with known AD-risk pathways that are relevant to the innate immune pathways, TREM2-TYROBP and IL-1β/TNFα, respectively. One gene, *TMEFF2*, is connected to the MTOR-APP pathway. From the top 20 VIP features (Table 1), several PPIs point to inflammatory processes and oxidative stress. Although limited by input data that do not include non-human (e.g., pathogen) information, this specific MPxgb(AD) model suggests *infection* as a key process in AD pathology[87–91]. Surprisingly, none of the LINCS/no-LINCS models generated in 2021 (LASSO or XGBoost) as opposed to the 2018 MPxgb(AD) model, ranked these five genes among the top 100 by probability (Supplementary Fig. 1b). This does not preclude the possibility that some of these top-ranked genes are relevant in AD pathology, and additional studies are needed.

While our model top and bottom predicted genes were experimentally screened, we note the approach has additional limitations. When using 3,952 negative label genes from OMIM, we assumed that association with other pathologies implies little or no role in AD/ADRD; however, this does not imply true negatives. Effective ML models require true negatives. In biology, this is not always trivial. Some genes (not directly related to immune mechanism) may indirectly influence immunity (e.g., pure metabolic genes may likely influence innate immune

function; likewise, genes in cell cycle regulation may affect immune cell proliferation/trafficking). We also limited ourselves with our positive labeled set, which had an immune network bias. As more established AD-associated and nonAD-associated genes become available, we anticipate improving the MPxgb(AD) model. Additional improvements may require parallel rankings derived from deploying different algorithms.

Two options might lead to improved prediction accuracy for absolute AD-associated vs. non-AD-associated genes. First, using positive-unlabeled learning to overcome the issue of noisy unlabeled data as negative samples[92,93], could lead towards a gradient distribution model. Indeed, unary/one-class classifiers trained on positive samples only exist[94]. Second, the use of true positives and true negatives and potentially more data types should be considered, e.g, from AD-specific databases with more multimodal attributes (e.g., disease-disease associations, protein domain, and sequence information). Based on the confusion matrix (Supplementary Table 3), we expected one in three positively predicted genes (up to 7 of the top 20) to be confirmed. We identified two strongly AD-associated proteins, with another three showing moderate association (Fig. 6b). Another limitation of this model is that MPxgb(AD) probability rank does not relate to a gradient in AD-relevance. Rather, the model was trained and weighted for predictions of potential AD-gene associations relevant in inflammatory pathways.

We also note a few caveats within our biological screening pipelines. While most of the genes were detectable in human iPSNs and autopsy brain samples, some genes (SLC44A2 and SCGB3A1) were undetectable in human brain samples via Western blot analysis. It is possible that the commercial antibodies utilized for these gene targets may not be ideal for Western blot analysis. Especially for the detection of SCGB3A1 in AD versus control samples, the use of a well-validated and reliable antibody would prove an even higher rank for potentiality for AD association. On the same note, in human AD autopsy brain samples compared to controls, many genes' mRNA levels were not significantly altered, yet most of the proteins were significantly altered in those same samples. These results conceivably suggest that increase in protein, but not mRNA, levels may occur post-transcriptionally during the progression of the disease. Furthermore, we used siRNA knockdowns to assess potential changes in an in vitro neuroinflammation mechanistic model measuring the effects of specific phosphorylation sites of tau. Most of the genes appear to play a significant role in the tau pathology within AD. However, it is essential to note that more extensive analyses need to be performed to detect off-target events. We also acknowledge these screening models exclude amyloid-β status, synaptic function, and other AD-relevant phenotypes. Additionally, this study did not examine the status of possible variants and their potential AD role within each target. Though some genes are expressed in the CNS, i.e., *NDRG2, GRINC2, TMEFF2, LMO4,* and *SLC44A2*[83,84,95–99], are top ranked genes previously shown to have little to unknown expression in the brain[50]. Moreover, some of our predicted AD-associated genes appear to have known AD associations (*NDRG2, TMEFF2, GRINC2*, and *LMO4*; see Supplementary Note 1). These studies, however, were more suggestive than confirmatory.

Despite these discrepancies, results from this study contribute to AD-associated gene knowledge by identifying potential AD-risk candidate genes from the top 20 MPxgb(AD) prediction list. These should be independently verified in future studies. Our three-pronged experimental screening system further contributes to the illumination of the complex AD network and the identification of potentially potential AD drug targets. We designed a unique method to characterize the potential role these genes may contribute to AD, more so, their role in tau pathology and with relevance to inflammation. We have shown in several ways that

*FRRS1, CTRAM, SCGB3A1, FAM92B/CIBAR2,* and *TMEFF2* are convincing candidates to further investigate as potential drug targets for AD.

In conclusion, we developed a specific MPxgb(AD) model to mine and predict AD-associated risk gene(s). Four of the twenty predicted genes were altered considerably at the mRNA level in sporadic AD iPSNs compared to control iPSNs. Nine of the twenty predicted genes have higher protein expression in AD brain tissue versus controls, and five have significantly upregulated mRNA levels. Finally, siRNA-mediated knockdown assays identified nine out of twenty genes that significantly reduce AD-relevant tau hyperphosphorylation at the AT8 and AT180 sites. Ranking the results based on the statistical significance of each experimental model, we identified *FRRS1* and *CTRAM* as having the most AD relevance within our AD model screening pipeline. Overall, these results provide a means to synthesize association data across studies likely to contribute to AD pathology. More testing and optimization in independent samples will warrant application in both future clinical practice and clinical trials.

## Methods

**ProteinGraphML build**. Briefly, when designing the MPxgb(AD), we first used the R version of ProteinGraphML program to build the PKG. We note that the current ProteinGraphML program is written in python. The ProteinGraphML software is designed to predict disease-to-protein (protein-coding gene) associations, from a biomedical knowledge graph via ML. This codebase abstracts the ML from the domain knowledge and data sources, to allow reuse for other applications. The input MySQL relational database is converted to a knowledge graph, then converted to feature vectors by metapath matching, based on an input disease, defining a training set of proteins. Then XGBoost[38] is used to generate and optimize a predictive model. We trained the model with an AD-focused gene set (Supplementary Table 2) and produced an AD-associated prediction list (Table 2), MPxgb(AD).

For further detail, data collected and integration of over 13 biological datasets across the major knowledge domains, including genomic, proteomic, functional, phenotypic, and interactions (biochemical, protein–protein, TF-DNA, etc.) into a PKG shown in Supplementary Table 1. Integration of multiple data types into the PKG enables, via the metapath approach, the formalization of different network paths connecting proteins to diseases (phenotypes). We implemented metapaths via SQL queries that extract matching metapath entities from TCRD to build an in-memory knowledge graph. Metapath examples are shown in summary form in Fig. 1a and listed with the corresponding SQL in Supplementary Table 6. We systematically transform the tabular result from each SQL into relationships between nodes. In addition to the meta path-based features, a set of static features is generated for sources invariant with disease query: GTEx[50], LINCS[49], CCLE[100], and HPA[101]. The SQL corresponding to these four feature sources is listed in Supplementary Table 7. Note, static features are *not* dependent on training set labels, only the database, so the same TSV files can be reused for all models, and only needs to be re-run if the database changes.

Graph analytics facilitates the computation of topological features such as path counts by enumerating all path instances matching a given metapath. Degree weighted path counts (*DWPCs*—see Eq. 1)[37] uses node-degree (number of edges) to weight each path instance and down-weight paths through higher-degree nodes. *DWPCs* quantify metapath prevalence using a dampening exponent ($w$, set to 0.4) to down-weight paths through high-connectivity nodes when computing *DWPCs* Eq. 1. The path-degree product $\Pi$ (Fig. 1b) is calculated by: (1) extracting all edge-specific degrees along the path $D_{path}$, where each edge contributes two degrees; (2) raising each degree $d$ to the $-w$ power, where $w$ is the dampening exponent; (3) multiplying all exponentiated degrees to yield $\Pi$. DWPC is the sum of path degree products $\Pi$ (adapted from ref.[37]). The metapath approach enables the transformation of the heterogeneous knowledge graph into an ML-ready input table, with one row per protein and one column per feature vector; features can be categorical or continuous variables. This transformation is a form of graph embedding, which refers to embedding into the feature vector space. Moreover, the PKG is transformed by matching the meta paths to the training set for a given input query and training set of known disease-associated genes. The static features are non-*DWPC* and not KG-based, and in this regard, we use a hybrid ML approach, whereas the approach by Himmelstein et al[37]. include non-*DWPC* features that are KG and path count based. Static features are *not* dependent on training set labels, only the database, so the same TSV files can be reused for all models, and only need to be re-run if the database changes. For machine learning, we selected XGBoost[38], an ML algorithm more rigorous than LightGBM[102]. XGBoost is known to be high performing, versatile, and amenable to interpretability, particularly relative to artificial neural networks and deep learning. XGBoost feature-importance scoring facilitates the ranking of meta paths for prioritization of manual follow-up to elucidate mechanistic hypotheses. While learning a classification model from the given data, the importance of features is calculated by XGBoost using three different metrics: gain, cover, and weight/

frequency. Gain implies the magnitude of the contribution of a particular feature to the model. The cover implies the average number of observations in which the feature was used to split the data across all trees in the model. Weight/frequency indicates how many times a feature was used to split the data across all trees in the model. Since the gain values represent the contribution of features, we used them to rank features. Gain values for features were calculated using the *get_score()* function of the XGBoost package[38]. The combination of metapath/XGBoost (MPxgb) processes assertions/evidence chains of heterogeneous biological data types and identifies similar assertions. The overall MPxgb algorithm and workflow are depicted schematically in Fig. 1a. For further description and tutorial description please visit https://github.com/unmtransinfo/ProteinGraphML or http://proteingraph.ml/

**Gene ontology enrichment analysis**. We performed GO enrichment analysis to examine whether predicted top genes were clustered into specific biological processes versus the bottom or least predicted genes. We used the enrichment analysis visualizer, https://appyters.maayanlab.cloud/#/Enrichment_Analysis_Visualizer and here: https://github.com/MaayanLab/appyter-catalog/tree/master/appyters/Enrichment_Analysis_Visualizer. This appyter uses four scores to report enrichment results: *p-value, q-value, rank (Z-score),* and *combined score*. The *p*-value is computed using Fisher's exact test or the hypergeometric test. This is a binomial proportion test that assumes a binomial distribution and independence for the probability of any gene belonging to any set. The *q*-value is an adjusted *p*-value using the Benjamini–Hochberg method for correction for multiple hypothesis testing. The rank score or z-score is computed using a modification to Fisher's exact test in which we compute a z-score for deviation from an expected rank.

**Human brain tissue samples**. Clinically and neuropathologically diagnosed human healthy control (HC) and Alzheimer's disease (AD) brain tissue samples were kindly provided by Northwestern Cognitive Neurology & Alzheimer's Disease Center (CNADC) Neuropathology Core. Details of the human samples used are provided in Supplementary Table 4. The University of New Mexico Institutional Review Board approved the use of all human autopsy specimens under exempt status. Informed consent was also obtained by all human subjects used in this study by the Northwestern University CNADC Neuropathology core, which provided the postmortem brain tissues for the study.

**Cell culture**. Human Neuroblastoma (SH-SY5Y (ATCC® CRL-2266) cells were maintained in Dulbecco's Modified Eagle Medium (Thermo Fisher Scientific) supplemented with 10% FBS (Thermo Fisher Scientific) and 100X Penicillin-Streptomycin (Thermo Fisher Scientific) at 37 °C. For the conditioned media (CM) experiment, LPS (1 μg/ml) was used to prime BV2 microglial cells. The BV2 microglial cells were a gift from Dr. Gary Landreth and were maintained in Dulbecco's Modified Eagle Medium (Thermo Fisher Scientific) supplemented with 10% FBS (Thermo Fisher Scientific) and 100X Penicillin-Streptomycin (Thermo Fisher Scientific). All cell lines tested negative for mycoplasma contamination using the Lonza MycoAlertTM Mycoplasma Detection Kit, Catalog #: LT07-118.

**Knock-down of gene of interest with siRNAs**. siRNAs used in both western blotting (WB) or immunohistochemistry (IHC) are listed in Supplementary Table 8. For the time-course kinetic experiment, the SH-SY5Y cells and BV2 cells were maintained and set up as above in cell culture methods. siRNA-mediated knockdown of target genes was performed using RNAiMax from Thermofisher manufacturer protocol (Cat. # 13778075). Briefly, SH-SY5Y cells were seeded ~$0.25^{-1} \times 10^6$ in a six-well plate. 150 μL of Opti-MEM Medium is mixed with 9 μL of Lipofectamine RNAiMAX reagent, then 150uL of Opti-MEM Medium is mixed with 30 pmol of individual siRNA against the top 20 genes (in alphabetical order; AKNA, BC02, CCNY, CRTAM, FAM92B, FOXP4, FRRS1, GRIN2C, 1L17REL, LILRA3, LM04, NDRG2, PIBF1, RAB40A, SCGB3A1, SLC44A2, SPOP, STARD3, TMEFF2, and TXNDC12). These two mixtures were then mixed 1:1 ratio together and incubated for 5 min at room temp. Then siRNA-lipid complex is added to cells (a single well). Simultaneously, BV2 cells were separately stimulated with lipopolysaccharide (LPS, 100 ng ml⁻¹). After 24 h at 37 °C, 50% of the siRNA-transfected SH-SY5Y media was replaced with LPS-stimulated BV2 conditioned media (CM). After another 24 h. at 37 °C, cells were lysed to detect phosphorylated tau and knockdown of GOIs. Confirmation was performed by SDS-PAGE and Western blot and by immunocytochemistry with antibodies against all twenty target proteins, as described below.

**Human-induced pluripotent stem cells**. Following inducible pluripotent stem cell lines (iPSC) were utilized: *sAD2.1*; Coriell # GM24666, (iPSCs derived from sporadic Alzheimer's disease patient; 83-year-old male), *control line*: Axol Bio # AX0018 (iPSC-Derived Neural Stem Cells; 74-year-old male). Briefly, iPSCs were maintained in mTESR plus supplement (STEMCELL Cat. No. 85850) at 37 °C. Neuron differentiation was performed as per the StemCell's neuron differentiation kit/protocol; (STEMCELL Cat. No. 05835, 05833, 08500, 08510). Later the medium was changed to BrainPhys™ (STEMCELL Cat. No. 05791) for long-term maturation at 37 °C. Neural progenitor cells seeded at $1.5 \times 10^4$ cells/cm² for maturation. Human cortical neurons supplements: $1 \times$ N2-Supplement A (STEMCELL Cat. No. 07152), $1\times$ NeuroCult™ SM1 without vitamin A (STEMCELL Cat. No. 05731), 200 nM ascorbic acid (Sigma Cat. No. A4403), 20 or 10 ng/ml

BDNF (STEMCELL Cat. No. 78133.1), 20 or 10 ng/ml GDNF (STEMCELL Cat. No. 78139.1), 1 μg/ml laminin (Thermo Fisher Scientific Cat. No. 23017015), and 0.5 or 0.25 mM dibutyryl cyclic-AMP (Sigma Cat. No. D0627).

**Gene expression analysis**. RNA from cells was extracted using the TriZOL reagent as described by the manufacturer (Thermo Fisher Scientific). Total RNA (20 ng/μL) was converted to cDNA using the High-Capacity cDNA Reverse Transcription kit (Thermo Fisher Scientific) and amplified using specific TaqMan assays (catalog # 4331182; Thermo Fisher Scientific). 18 s rRNA (catalog # 4319413E, Thermo Fisher Scientific) was used as a housekeeping gene for normalization. The list of qRT-PCR assays are listed in Supplementary Table 9 and were run on the StepOnePlus® Real-Time PCR System (Thermo Fisher Scientific) and the statistical analyses were performed using Prism.

**SDS-PAGE and western immunoblotting**. Cells were lysed in 1x NuPAGE LDS Sample Buffer (Thermo Fisher Scientific Cat. No. NP0007) and NuPage Sample Reducing Agent (Thermo Fisher Scientific Cat. No. NP0009) and sonicated for 30 s, boiled at 95 °C for 15 min. Cell lysates were resolved via SDS-PAGE and immunoblotted as previously described (5). The dilutions of primary antibodies utilized are indicated in Supplementary Table 10. For the detection of phosphorylated tau, AT8 (Thermo Fisher Scientific Cat. No. MN1020AT8; for tau phosphorylated at S199/S202) and AT180 (Thermo Fisher Scientific Cat. No. MN1040; for tau phosphorylated at T231) monoclonal antibodies were utilized. Anti-actin antibody was used as loading control. For the human autopsy brain samples, temporal cortices were homogenized in 10% weight/volume Tissue Protein Extraction Reagent (T-PER®, Thermo Fisher Scientific) with protease (P8340 Sigma-Aldrich) and phosphatase (P5726; Sigma-Aldrich) inhibitor cocktails. Protein (20 μg) was resolved in 4–12% Bis-Tris Novex NuPage gels (Invitrogen) and transferred to PVDF membrane, blocked (in 5% milk), and incubated overnight in primary antibodies (details on the dilutions are provided in Supplementary Table 10 followed by incubations with respective secondary antibodies. Membranes were developed using ECL reagent (NEL101001EA; Perkin Elmer) and immunoreactive bands were quantified in AlphaEaseFC™ Software (Alpha Innotech Corporation).

**Immunocytochemistry and confocal microscopy**. Cells were plated on coverslips coated with laminin, once cells were ready for fixation, they were fixed in 4% paraformaldehyde, blocked with 0.2% Triton and 10% donkey serum, incubated in primary overnight in 4 °C (5% donkey serum). AT8, AT180 anti-Tubulin β-3 (TUBB3 Antibody, BioLegend Cat. # 802001) at 1:250, were utilized as primary antibodies (Supplementary Table 10). Goat anti-mouse Alexa Fluor 488 and Alexa Fluor 555 (1:500) secondary antibodies were used. DAPI was used to stain the nucleus. Coverslips were mounted on slides using Fluoromount-G (ThermoFisher, Cat# 00-4958-02). Immunofluorescence confocal microscopy was carried out using Zeiss LSM 510 Meta microscope with ZEISS ZEN imaging Software. To confirm the knockdown of target genes via immunocytochemistry, four random fields per immunostained epifluorescence images were quantified for the average immunoreactive area per field using Image J software. Three technical replicates from three different biological replicates per condition were imaged and quantified. Mean fluorescence intensity (MFI) across different replicates per condition was compared and plotted. Any values that fell either above or below mean + 2*SD were called out as outliers and excluded from the analysis.

**Antibodies**. Antibodies used in western blotting (WB) or immunohistochemistry (IHC) are listed in Supplementary Table 10.

**Statistics and reproducibility**. Unless otherwise indicated, comparisons between two groups were done via unpaired two-tailed t-test with Welch's correction; comparisons between multiple treatment groups were done via one-way or two-way analysis of variance (ANOVA) with indicated multiple comparisons post hoc tests. Tukey's method was used to find and remove outliers. All statistical analyses were performed using GraphPad Prism® (Version 9.0). For iPSN and siRNA experiments, $n = 3$ biological replicates; $n = 3$ technical replicates were used.

**Reporting summary**. Further information on research design is available in the Nature Research Reporting Summary linked to this article.

## Data availability
Data used for analysis in this study and raw images are available at https://figshare.com/projects/Machine_learning_prediction_and_tau-based_screening_identifies_potential_Alzheimer_s_disease_genes_relevant_to_immunity/127145. All other data are available from the corresponding author upon reasonable request.

## Code availability
We used version 6 of TCRD and can be downloaded at http://juniper.health.unm.edu/tcrd/download/. We used the R version of ProteinGraphML program to build the PKG.

We note that the current ProteinGraphML program is written in python and is available at https://github.com/unmtransinfo/ProteinGraphML. For full analysis are at https://doi.org/10.5281/zenodo.5784581.

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

## Acknowledgements

This work was primarily funded by the NIH Common Fund U24 CA224370-01S1 AD/ADRD supplement. Additional funding for this study was from RF1NS083704-05A1, R01NS083704, R21NS077089, and R21NS093442; UNM Health Sciences Center Bridge Funding, UNM Department of Molecular Genetics and Microbiology intradepartmental grant funding, Dr. Stephanie Ruby travel award (to J.B.). This study was also supported in part by an Alzheimer's Disease Core Center grant (P30 AG013854) from the National Institute on Aging to Northwestern University, Chicago Illinois. We gratefully acknowledge the assistance of the Northwestern Cognitive Neurology & Alzheimer's Disease Center (CNADC) Neuropathology Core for postmortem tissue samples. The general machine learning, informatics, and data science framework development was supported by the IDG KMC application from the University of New Mexico (NIH CA224370). Additional funding for T.I.O., C.B., and J.J.Y. was provided by NIH grant U24 TR002278.

## Author contributions

T.I.O., K.B., J.B., C.B., and O.U. designed and planned all the experiments. O.U., T.I.O., and C.B aggregated data and developed machine learning models. O.U., P.K., and J.J.Y. validated the machine learning and data collection framework. J.B., N.M., S.D., D.S., and S.J. performed all the screening experiments. J.W. provided iPSCs. O.M. performed statistical analysis. J.B., T.I.O., and K.B. wrote the manuscript and prepared all the datasets.

## Competing interests

T.I.O. has received honoraria or consulted for Abbott, AstraZeneca, Chiron, Genentech, Infinity Pharmaceuticals, Merz Pharmaceuticals, Merck Darmstadt, Mitsubishi Tanabe, Novartis, Ono Pharmaceuticals, Pfizer, Roche, Roivant Discovery, Sanofi, and Wyeth. He is on the Scientific Advisory Board of ChemDiv Inc. and InSilico Medicine. The remaining authors declare no competing interests.
