## [Peer Review File · Communications Biology]

Reviewers' comments:

Reviewer #1 (Remarks to the Author):

The authors report the development of an AD-specific machine learning model (MPxgb) and its application for the successful identification of novel AD-associated genes. They used data aggregated in the TargetCentral Resource Database and collated 13 distinct datasets, totaling over 261 million attributes, covering several protein attributes. MPxgb was used to predict genes based on their ability to alter tau pathology in three different experimental systems: iPSNs from sporadic AD patients, AD brain samples and a SHSY5Y neuronal cell culture model of inflammation-induced tau pathology. MPxgb predicted twenty genes involved in various pathways including immune function. mRNA levels of eleven of the twenty predicted genes were significantly altered in sporadic AD iPSNs compared to controls iPSNs. Additionally, nine and five of the twenty predicted proteins were upregulated in AD brains and iPSNs, respectively. siRNA knockdown of the top seven genes in SHSY5Y cultures significantly reduced pathologic tau phosphorylation on Ser199/Ser202 and Thr231 sites. Finally, three (PIBF1, LILRA3 and CRTAM) of the top seven genes showed the most significant effect on tau phosphorylation and two (CRTAM and LILRA3) novel genes were found to be in the TREM2-TYROBP pathway implicated in innate immunity.

The authors have designed an interesting machine learning/artificial intelligence model to help identify genes that are involved in AD pathogenesis. As such, their model represents a very useful tool for the field, which is expected to help advance the identification of novel therapeutic targets for this devastating neurodegenerative disease.

Reviewer #2 (Remarks to the Author):

The authors present a study that aims to identify genes that influence tau pathology in Alzheimer's disease (AD). They accomplish this by applying a previously published machine learning framework to an AD-specific protein knowledge graph that integrates a multitude of datasets encompassing different data types.

Among the features most strongly associated with AD are pathways related to neuroinflammation, such as oxidative stress, gene sets associated with cytokines and the JAK/STAT-pathway. This finding is consistent with the increasingly prevalent hypothesis that neuroinflammation plays a central role in AD pathology, see e.g. [https://doi.org/10.1016/S1474-4422\(15\)70016-5](https://doi.org/10.1016/S1474-4422(15)70016-5)

From these features the authors extract a number of candidate genes whose impact on tau pathology is then validated in three AD models using RNAi. The validated candidates are of wide interest as targets for the development of disease modifying compounds.

All in all, the manuscript is novel and important, contributing an interesting set of potentially disease modifying genes. However, in my opinion, there are three issues with the current manuscript that need clarification:

1. The generation of the knowledge graph and the machine learning framework is not detailed in the methods at all. There is some information in the main text, but not sufficient to reproduce the results. If parts of the original meta-path method were modified then these changes should be summarized in the methods. The code is available on Github, but information necessary for reproduction, e.g. the hyperparameters that were used are missing.
2. It is unclear how the individual gene hits are derived from the MPxgb model features (line 224). For example, I had trouble following the argument line 226. Does that mean that AKNA is consistently a member of the gene sets (MPxgb model features) at the top of the VIP list?
3. The validation of gene hits relies on RNAi knockdown, which has proven to be prone to false positive effects due to off-target effects. Did the authors take any steps to mitigate the frequent off-target effects of RNAi? Off-target effects of RNAi, especially siRNA, are very common and lead

to misleading results. See e.g. <https://doi.org/10.1371/journal.pbio.2003213>

Furthermore, I have a number of additional comments:

1. The abstract, and also the manuscript in general, requires some editing. There are grammatically ambiguous or incomplete sentences that sometimes make the manuscript difficult to read - too many to list. E.g. in the abstract, It is unclear what "(PKG)/meta-path (m-p)/ML" and "MPxgb" refer to and the sentence "Second, mRNA levels of eleven of the twenty predicted genes significantly altered in the sporadic AD iPSNs compared to controls iPSNs" is incomplete.
2. The properties of the protein knowledge graph should be explained in more detail. What does the number of direct edges between the same two given nodes represent? Different databases providing the same information or multiple sources of evidence (publications?) from a single database? Can the number of edges between two nodes be used as a proxy for the confidence in that interaction?
3. Some statements in the discussion should be clarified. E.g. "Various isoforms or subtypes of these genes seem to delegate towards novel glial or neuronal functions related to AD" (what is delegated? line 421) and "Altogether, the novel genes identified in our study do not necessarily correlate with each other" (In what way don't they correlate? line 419).
4. The authors could consider representing the information in Fig. 8A summarizing the performance of each candidate gene in the validation assays as heatmap or table instead. The meaning of each ring is rather difficult to gauge at the moment.

Reviewer #3 (Remarks to the Author):

This paper used a machine learning algorithm as a hypothesis generating tool, to identify genes that may be associated with Alzheimer's Disease. Twenty of the genes that were predicted to be associated with AD were selected for further analysis, and a series of experiments were carried out to assess their degree of association. These experiments included differential expression analyses in cell models, differential expression analyses from post-mortem brain tissue, and a study of the effects of siRNA knockdowns on tau phosphorylation in cell models. At the conclusion of these experiments, the authors claim to have identified 3 genes that are strongly associated with AD pathology, and 4 that are moderately associated; some of these genes had not been associated with AD in previous studies, but others had.

At a high level, the methodology applied in the paper (to the extent I could follow it, more on that below) seems reasonable. Identifying some novel genes that may be associated with AD pathology is certainly interesting, and further study of these genes is merited.

My major comments on the manuscript fall into two categories:

1. Parts of this paper are nearly incomprehensible. Some information that is presented seems completely unnecessary; for example, what does Table 1 add to this manuscript? What information is the reader supposed to get from Figure 1? More importantly, the description of the machine learning part of the study is very difficult to follow. In the end, a gradient boosted tree classifier (such as one implemented in the software library xgboost used in the paper) learns a function $f: x \rightarrow y$, in which x is a vector of features and y is a label. The feature vectors (x) were somehow computed from a knowledge graph, but the description of that process is so convoluted that I could not follow it. Ultimately, I don't really know what x was and, as a result, can't really comment on whether or not I think the method was appropriate. I think that this section needs to be rewritten, with simple clear descriptions of f (it's not enough to just cite xgboost), x , and y .
2. The authors don't include any negative control genes in their experiments. That is, they take the top 20 genes that were predicted to be associated with AD and use them for their experiments. However, in order to judge whether or not the ML algorithm was ultimately useful,

I'd like to see similar experiments on 20 genes that were predicted to NOT be associated with AD. If those genes also ended up being differentially expressed, or their knockdowns had effects on tau phosphorylation, then what would that imply about the utility of the ML algorithm? Maybe any 20 randomly chosen genes would have the same experimental signals? Without showing these analyses, the authors can't really claim that their ML algorithm was useful.

I also have a few minor comments:

1. The authors state that the variable importance ranking of a gradient boosted tree classifier lead to "directly interpretable mechanistic insights." This is simply not true. A gradient boosted tree is not performing causal inference, and simply identifies (nonlinear) correlations.
2. The authors state multiple times that AI, ML and "accurate data science" are needed to solve some problems. I found this statement to be very weird, and could not understand how they are defining these terms.
3. The figure legends are generally insufficient for understanding the figures (e.g, Figure 2). Figure 5 is mislabeled it has parts a, b, d, and e but no part c.

Response to the Editorial Office:

Question (1) Elaborate on the specifics of the machine learning model, as suggested by Referees #2-3.

Response (1): We have completely rewritten the MPxgb(AD) model description and redesigned the MPxgb(AD) figure.

Question (2) Validate the "least associated" targets from your machine learning approach in vitro, as suggested by Referee #3. We believe that 10-20 targets would be sufficient for the purpose of validation in each paradigm.

Response (2): We ran the bottom 10 genes through the first two screening AD models. From doing these analyses we ran a more strenuous statistical method across all data sets. Comparisons between two groups (AD vs healthy controls) were redone via unpaired two-tailed t test with Welch's correction. This changed the significance levels for multiple assays. Originally our top three genes were PIBF1, LILRA3, and CRTAM. After new statistical analysis and elimination of the siRNA data points, FRRS1, CTRAM, SCGB3A1, FAM92B/CIBAR2, and TMEFF2 are the top gene hits.

Question (3) Carefully proofread text and figures to improve readability.

Response (3): We appreciate the overall concern and agreed that the manuscript needed great due diligence, we significantly rewrote and fixed any grammatical errors.

Responses to Reviewers:

Reviewer #1

The authors have designed an interesting machine learning/artificial intelligence model to help identify genes that are involved in AD pathogenesis. As such, their model represents a very useful tool for the field, which is expected to help advance the identification of novel therapeutic targets for this devastating neurodegenerative disease.

Reviewer #2

Question (2-1). The generation of the knowledge graph and the machine learning framework is not detailed in the methods at all. There is some information in the main text, but not sufficient to reproduce the results. If parts of the original meta-path method were modified then these changes should be summarized in the methods. The code is available on Github, but information necessary for reproduction, e.g. the hyperparameters that were used are missing.

Response (2-1): We link to the ProteinGraphML web app as well as github. We also state that we no longer have access to that specific (Feb 2018) RGD download. We no longer use the R code, and instead have migrated to a python code. This does not preclude the validity of the predictions, nor the novel genes discovered via MPxgb(AD).

Question (2-2). It is unclear how the individual gene hits are derived from the MPxgb model features (line 224). For example, I had trouble following the argument line 226. Does that mean that AKNA is consistently a member of the gene sets (MPxgb model features) at the top of the VIP list?

Response (2-2): We addressed this, by significantly redescribing the MPxgb(AD).

Question (2-3). The validation of gene hits relies on RNAi knockdown, which has proven to be prone to false positive effects due to off-target effects. Did the authors take any steps to mitigate the frequent off-target effects of RNAi? Off-target effects of RNAi, especially siRNA, are very common and lead to misleading results. See e.g. <https://doi.org/10.1371/journal.pbio.2003213>

Response (2-3): We agree that siRNA experiments may have potential off-target effects. We have added language that mentions the cons/caveats to this approach and acknowledge further experiments are necessary. We further omitted siRNA experiments from our overall ranking score for evaluating the predicted gene list. Therefore we did not run the bottom 10 genes through the siRNA screening method.

Question (2-4). The abstract, and also the manuscript in general, requires some editing. There are grammatically ambiguous or incomplete sentences that sometimes make the manuscript difficult to read - too many to list. E.g. in the abstract, It is unclear what “(PKG)/meta-path (m-p)/ML“ and “MPxgb” refer to and the sentence “Second, mRNA levels of eleven of the twenty predicted genes significantly altered in the sporadic AD iPSNs compared to controls iPSNs” is incomplete.

Response (2-4): We appreciate Reviewer #2’s concerns and agreed that the manuscript needed great due diligence. We significantly rewrote and fixed any grammatical errors.

Question (2-5). The properties of the protein knowledge graph should be explained in more detail. What does the number of direct edges between the same two given nodes represent? Different databases providing the same information or multiple sources of evidence (publications?) from a single database? Can the number of edges between two nodes be used as a proxy for the confidence in that interaction?

Response (2-5): The Methods section now contains more detailed information on how the protein knowledge graph is built. There is only one edge between any 2 nodes in PKG. However a protein might be connected through multiple intermediate nodes to a concept like function or disease, in this case the multiple intermediate nodes represent information from multiple sources (database, publications, etc.) and are used to compute weighted path counts DWPCs (degree weighted path counts) that are a proxy for biological context associated with protein - disease pair.

In a well designed KG, entities and relationships would be integrated with rigor to avoid duplication. Two databases leading to the same semantic triple (subject-[relationship]->object) would then not result in a double count. ProteinGraphML uses DWPCs to downweight highly connected nodes, etc. The 13 sets of data integrated in this model provide non-overlapping (for the most part) sets of evidence. DWPCs serve as proxy for confidence, and each meta path based feature reflects such a DWPC-based confidence.

Question (2-6). Some statements in the discussion should be clarified. E.g. “Various isoforms or subtypes of these genes seem to delegate towards novel glial or neuronal functions related to AD” (what is delegated? line 421) and “Altogether, the novel genes identified in our study do not necessarily correlate with each other” (In what way don’t they correlate? line 419).

Response (1): We fixed or omitted these sentences.

Question (2-7). The authors could consider representing the information in Fig. 8A summarizing the performance of each candidate gene in the validation assays as heatmap or table instead. The meaning of each ring is rather difficult to gauge at the moment.

Response (2-7): We changed the performance graph to a heatmap for better visualization.

Reviewer #3

Question (3-1). Parts of this paper are nearly incomprehensible. Some information that is presented seems completely unnecessary; for example, what does Table 1 add to this manuscript? What information is the reader supposed to get from Figure 1? More importantly, the description of the machine learning part of the study is very difficult to follow. In the end, a gradient boosted tree classifier (such as one implemented in the software library xgboost used in the paper) learns a function $f: x \rightarrow y$, in which x is a vector of features and y is a label. The feature vectors (x) were somehow computed from a knowledge graph, but the description of that process is so convoluted that I could not follow it. Ultimately, I don't really know what x was and, as a result, can't really comment on whether or not I think the method was appropriate. I think that this section needs to be rewritten, with simple clear descriptions of f (it's not enough to just cite xgboost), x , and y .

Response (3-1): We appreciate Reviewer #3's concerns and agreed that the manuscript needed great due diligence, we significantly rewrote and fixed any grammatical errors. The original (Figure 1) was eliminated. We also removed the original (Table 1) that listed current AD drugs. We rewrote, in better detail, our machine learning description. The feature vectors (x) were derived from the meta path algorithm. We described the top features in detail.

Question (3-2). The authors don't include any negative control genes in their experiments. That is, they take the top 20 genes that were predicted to be associated with AD and use them for their experiments. However, in order to judge whether or not the ML algorithm was ultimately useful, I'd like to see similar experiments on 20 genes that were predicted to NOT be associated with AD. If those genes also ended up being differentially expressed, or their knockdowns had effects on tau phosphorylation, then what would that imply about the utility of the ML algorithm? Maybe any 20 randomly chosen genes would have the same experimental signals? Without showing these analyses, the authors can't really claim that their ML algorithm was useful.

Response (3-2): We are extremely thankful for this suggestion. Upon analyzing the bottom 10 genes, we discovered that our ML model does not necessarily rank the predicted genes in a "most likely to be AD-associated to least likely or NOT AD-associated" manner. This forced us to perform in-depth analyses and cross validate the results in order to better understand MPxgb(AD) model output. We added a section to discuss the bottom ten genes, characterized via two of our biological screening

methods. We also elaborated on our model's weaknesses and strengths in the discussion.

Question (3-3). The authors state that the variable importance ranking of a gradient boosted tree classifier lead to "directly interpretable mechanistic insights." This is simply not true. A gradient boosted tree is not performing causal inference, and simply identifies (nonlinear) correlations.

Response (3-3): This is absolutely true, and we agree 100%, we fixed this statement!

Question (3-4). The authors state multiple times that AI, ML and "accurate data science" are needed to solve some problems. I found this statement to be very weird, and could not understand how they are defining these terms.

Response (3-4): We agree and removed this type of language.

Question (3-5). The figure legends are generally insufficient for understanding the figures (e.g, Figure 2). Figure 5 is mislabeled; it has parts a, b, d, and e but no part c.

Response (3-5): We updated and edited all figures.

Reviewers' comments:

Reviewer #2 (Remarks to the Author):

I want to thank the authors for their revised manuscript. The revisions helped a lot to clarify how the knowledge graph was assembled and many aspects of the machine learning model setup. The comparison between the 20 highest ranked genes and the 10 lowest ranked genes as predicted by the MPxgb(AD) model is a great addition and helps interpretation of the model predictions.

I still have some concerns about the description of the model. Reading the manuscript I still often had to refer to reference 37 (<https://doi.org/10.1371/journal.pcbi.1004259>) in order to understand the method.

For example, the authors don't specify how they applied the DWPC method in their work. Reference 37 includes the statement "We calculated DWPC features for the 22 metapaths of length 3 or less that originated with a gene and terminated with disease. Two non-DWPC features were included to assess the pleiotropy of the source gene and the polygenicity of the target disease". Did the authors use the same approach?

How feature importance was calculated is also not specified. Same as in reference 37 ("We adopt standardized coefficients as a measure of feature effect size. Standardized coefficients refer to the coefficients from logistic regression when all features have been transformed to z-scores. Standardization provides a common scale to assess feature effect, both within and across models.")?

Reviewer #3 (Remarks to the Author):

I thank the authors for making an effort to address the reviewers' concerns, and was very glad to find the open source project available on github. However, I still have questions on whether the model that has been developed is particularly useful.

1. The model has a very large number of features in comparison to the number training of samples. Therefore, I'd expect model performance to have a high variance. Indeed, a small change to the loss function during training (i.e., reweighting or resampling to counteract the class imbalance) leads to dramatically different performance characteristics out-of-sample. The authors choose the model that performed best on the test set, but by making a choice negate its use as an actual test set -- model selection is ultimately a type of model training. It would be more convincing to reproduce figures 1c,d with something like 10-fold cross-validation so that readers could assess the variability in the ROC curves. Is it just a fluke that the two loss functions gave such different performance on this test set? Maybe that's just how variable the ROC curves are?

2. I think the model has too many features and too few samples for a feature selection approach like this to be particularly meaningful. I say "I think" because it isn't clear to me exactly how many features were used in training the gradient boosted decision trees. In general, highly weighted features will often vary from one cross-validation fold to the another.

To see why, suppose we want to predict y from x_1, x_2, \dots, x_N , and that we use a feature selection algorithm. In addition, suppose x_1 is highly correlated with x_2 . Because x_1 and x_2 are highly correlated with each other, they must have very similar correlations with y . Whether x_1 or x_2 has the larger correlation with y will depend, randomly, on the choice of samples to include in the training set during cross-validation. So the feature importance rankings aren't stable.

Therefore, the above suggestion for 10-fold cross validation would also allow you to estimate the variance in the feature importance rankings.

3. In a typical machine learning paper, we would like to see more comparisons to understand what part of the features/model are really doing the work. For example, how would performance compare to a logistic regression with a LASSO penalty, or to an SVM? Would the feature

importance metrics be the same?

What about dropping some of the features sets? You say that LINCS is the largest category, but also the most selected. Well, is it most selected simply because it's the largest, so it has the most chances, or because it's actually important? You could test this just by deleting LINCS and comparing the results (with cross validation). Similar experiments with the other datasets would also be helpful for the same reason.

Responses to Reviewers:

Reviewer #2 (Remarks to the Author):

I want to thank the authors for their revised manuscript. The revisions helped a lot to clarify how the knowledge graph was assembled and many aspects of the machine learning model setup. The comparison between the 20 highest ranked genes and the 10 lowest ranked genes as predicted by the MPxgb(AD) model is a great addition and helps interpretation of the model predictions.

Question (2-1): I still have some concerns about the description of the model.

Reading the manuscript I still often had to refer to reference 37

(<https://doi.org/10.1371/journal.pcbi.1004259>) in order to understand the method.

For example, the authors don't specify how they applied the DWPC method in their work. Reference 37 includes the statement "We calculated DWPC features for the 22 metapaths of length 3 or less that originated with a gene and terminated with disease. Two non-DWPC features were included to assess the pleiotropy of the source gene and the polygenicity of the target disease". Did the authors use the same approach?

Response (2-1): We agree that the manuscript describes the static features and specific data sources (LINCS, GTEX, HPA, CCLE) in methods. We thank the Reviewer for pointing out that we did not clarify that these "static features" are non-DWPC and moreover not KG-based. In this regard our approach is hybrid, whereas Himmelstein et al. (Ref 37) include non-DWPC features which are KG and path count based. We have now added language to further describe the static features in the **Methods** section.

Moreover, the PKG is transformed by matching the meta paths to the training set for a given input query and training set of known disease associated genes. The static features are non-DWPC and not KG-based, and in this regard, we use a hybrid ML approach, whereas the approach by Himmelstein et al.³⁷ include non-DWPC features that are KG and path count based. Static features are *not* dependent on training set labels, only the database, so the same TSV files can be reused for all models, and only need to be re-run if the database changes.

Question (2-2): How feature importance was calculated is also not specified.

Same as in reference 37; "We adopt standardized coefficients as a measure of feature effect size. Standardized coefficients refer to the coefficients from logistic regression when all features have been transformed to z-scores. Standardization provides a common scale to assess feature effect, both within and across models."?

Response (2-2): We originally did not provide complete details regarding the XGBoost machine learning algorithm and how it determines the importance of features in generating gradient boosted trees with the minimized loss function. We have now added the following underlined language to the manuscript in the **Methods** section.

While learning a classification model from the given data, the importance of features is calculated by XGBoost using three different metrics: gain, cover, and weight/frequency. "Gain" implies the magnitude of the contribution of a particular feature to the model. "Cover" implies the average number of observations in which the feature was used to split the data across all trees in the model. "Weight/frequency" indicates how many times a feature was used to split the data across all trees in the model. Since the "gain" values represent the contribution of features, we used them to rank features. "Gain" values for features were calculated using the `get_score()` function of the XGBoost package³⁸.

<https://xgboost.readthedocs.io/en/latest/R-package/discoverYourData.html#feature-importance>: "**Gain** is the improvement in accuracy brought by a feature to the branches it is on. The idea is that before adding a new split on a feature X to the branch there was some wrongly classified elements, after adding the split on this feature, there are two new branches, and each of these branches is more accurate (one branch saying if your observation is on this branch then it should be classified as 1, and the other branch saying the exact opposite). **Cover** measures the relative quantity of observations concerned by a feature. **Frequency** is a simpler way to measure the Gain. It just counts the number of times a feature is used in all generated trees."

Reviewer #3 (Remarks to the Author):

I thank the authors for making an effort to address the reviewers' concerns and was very glad to find the open source project available on github. However, I still have questions on whether the model that has been developed is particularly useful.

(Question 3-1): The model has a very large number of features in comparison to the number training of samples. Therefore, I'd expect model performance to have a high variance. Indeed, a small change to the loss function during training (i.e., reweighting or resampling to counteract the class imbalance) leads to dramatically different performance characteristics out-of-sample. The authors choose the model that performed best on the test set, but by making a choice negate it's use as an actual test set -- model selection is ultimately a type of model training. It would be more convincing to reproduce figures 1c,d with something like 10-fold cross-validation so that readers could assess the variability in the ROC curves. Is it just a fluke that the two loss functions gave such different performance on this test set? Maybe that's just how variable the ROC curves are? I think the model has too many features and too few samples for a feature selection approach like this to be particularly meaningful. I say "I think" because it isn't clear to me exactly how many features were used in training the gradient boosted decision trees. In general, highly weighted features will often vary from one cross-validation fold to the another. To see why, suppose we want to predict y from x1, x2, ..., xN, and that we use a feature selection algorithm. In addition, suppose x1 is highly correlated with x2. Because x1 and x2 are highly correlated with each other, they must have very similar correlations with y.

Whether x1 or x2 has the larger correlation with y will depend, randomly, on the choice of samples to include in the training set during cross-validation. So the feature importance rankings aren't stable. Therefore, the above suggestion for 10-fold cross validation would also allow you to estimate the variance in the feature importance rankings.

Response (3-1): We have previously disclosed our inability to reproduce the exact 2018 model and its prediction results. The initial data files are from February 2018, and much has changed: all our databases and data sets have been updated; the seed number (a critical random starting point that influences results) as well as code versioning had not been documented - this was quite literally our first MPxgb model. To that extent, it is not possible to reproduce Figure 1c-d with a 10-fold cross-validation (CV) since that figure would no longer reflect the 2018 data. Back in 2018, we performed 5-fold CV. We ran 10-fold CV on our 2021 datasets and databases. Results were as follows:

The dataset we used to train the model had 4,005 samples (53 positives and 3,952 negatives) and 22,549 features. The gradient boosting tree method (XGBoost) estimates feature importance and uses important features to learn the model. Although the dataset had 22,549 features, only 692 features contributed to boosted tree building. Out of those 692, only 242 features had "gain" > 0.0001, and the remaining features had "gain" < 9.99e-05. The underlined text has now been incorporated into the manuscript in the ***Focus on Alzheimer's disease*** section.

Since the dataset is highly imbalanced (negative to positive ratio: 74.57), feature selection by any feature extraction method will be biased towards the majority class. Therefore, we did not use traditional feature extraction methods. Instead, we used the "scale_pos_weight" parameter of XGBoost to scale the positive class so that the model is not biased towards the majority negative class. We edited and added the underlined language to the manuscript in the ***Focus on Alzheimer's disease*** section. "Given the highly imbalanced nature of the training set (more negatives than positives), we addressed this issue in two ways: 1) assign higher weights to the AD-associated (positive label) genes using the "scale_pos_weight" parameter of XGBoost; or 2) generate a balanced training set by sampling with replacement of positives to match the number of negatives."

We appreciate the Reviewer's suggestion regarding ten-fold CV. As suggested, we ran the XGBoost model with 10-fold CV on the dataset. The underlined text is now part of the manuscript - see also **Supplementary Figure S7**. All comparative results are uploaded on GoogleDrive at this link: <https://bit.ly/2ZWnlvz>

Having not archived the seed (a random starting point for ML models that influences results) for the 2018 MPxgb(AD) model, we can no longer reproduce this model. It is however essential to re-assess datasets and databases with the same input/training sets over time to determine reproducibility and feature importance metrics between various models. Upon running a 10-fold CV for XGBoost on the 2021 version of the database (**Supplementary Fig. S7a**), AUC-ROC values ranged from 0.8120-0.9889 (mean: 0.8822). XGBoost used between 209 and 249 features out of 22,549 to learn the models. As model training selected different sets of records to train the model

in each fold of the 10-fold CV, the list of important features differed between models. The common features among ten models varied from 39 to 61. We used weighted models in the 10-fold CV, and the mean AUC-ROC is comparable to what we reported for the 2018 5-fold CV weighted models (**Fig. 1c**). Although there were 39-61 features common in 10 models, their rankings were not the same. Depending on the data selected to train the models in 10-fold CV, the contribution of features also varied. In comparison to XGBoost models, LASSO models have a considerable AUC-ROC variance (**Supplementary Fig. S7a**). Like XGBoost, LASSO regression automatically selects features to learn a model. It used 8-79 features out of 22,549 to learn the models in the 10-fold CV. The common features among ten models varied from 6-34.

We compared ML model performance by excluding LINCS features from our learning set. After dropping LINCS, the train and test datasets had 3,602 features. Without LINCS, XGBoost models ranged from 0.6023-0.9994 (mean: 0.8701). Comparing the performance of models with/without LINCS (**Supplementary Fig. S7a**), the lower bound of the AUC dropped, but the mean AUC and upper bound remained similar. XGBoost used 181-206 features out of 3,602 to learn the models. The common features among ten models varied from 56-82. After dropping LINCS, XGBoost models had more features in common. There was no significant change in the lower and upper bound of AUC when dropping LINCS in LASSO models, but the mean AUC increased, 0.6807-0.9977 (mean: 0.9085). LASSO used 34-446 features out of 3,602 to learn the models. The common features among ten models varied from 18-145. After dropping LINCS, LASSO comparatively used more features to learn models, and the models had more features in common. Among the top 20 features of LASSO and XGBoost models without LINCS, 2-4 features were the same. We did not observe significant changes in the count of common features among XGBoost and LASSO models after dropping LINCS. There was a decent overlap when examining the top 100 predicted genes among various models (**Supplementary Fig. S7b**). When including LINCS, XGBoost performs slightly better, whereas LASSO performs slightly better without LINCS. Given this retrospective evaluation (2021 vs 2018 models), and for the sake of interpretability, we continue to discuss the results from XGBoost as the algorithm of choice for our ML model.

(Question 3-2): In a typical machine learning paper, we would like to see more comparisons to understand what part of the features/model are really doing the work. For example, how would performance compare to a logistic regression with a LASSO penalty, or to an SVM? Would the feature importance metrics be the same? What about dropping some of the features sets? You say that LINCS is the largest category, but also the most selected. Well, is it most selected simply because it's the largest, so it has the most chances, or because it's actually important? You could test this just by deleting LINCS and comparing the results (with cross validation). Similar experiments with the other datasets would also be helpful for the same reason.

Response (3-2): We thank the Reviewer for suggesting other performance metrics to compare logistic regression. Unfortunately, using SVM was not practical, as the R library (e1071) we used for SVM does not support multithreading. The SVM code took several hours to finish just one iteration of the 10-fold CV. Its comparatively slow performance on this dataset was possibly compounded by the large number of features. In short, we did not compare our results with SVM. Our combined XGBoost / LASSO discussion has been included above (see the ***Focus on Alzheimer's disease*** section). Below we also include the Supplementary Figure 7.

Figure S7. Comparison of different cross validation models. a) AUC-ROC curves of XGBoost and LASSO with or without LINC selected. For XGBoost with LINC, the value of AUC-ROC ranged from 0.8120-0.9889 (mean: 0.8822) in the ten-fold CV. For LASSO regression to compare the performance of classifiers, the value of AUC-ROC for LASSO with LINC models varied from 0.6848-0.9966 (mean: 0.8397) in the 10-fold

CV. In comparison to XGBoost models, LASSO models have a considerable variance in AUC-ROC. For XGBoost without LINCS, the value of AUC-ROC ranged from 0.6023-0.9994 (mean: 0.8701) in the ten-fold CV. For LASSO without LINCS, the value of AUC-ROC ranged from 0.6807-0.9977 (mean: 0.9085) in the ten-fold CV. Overall with LINCS, XGBoost performs slightly better, and LASSO performs slightly better without LINCS. **b)** Using the “Compare Sets Appyter” (Clarke et al. 2021), we compare the various 2021 models and our 2018 model top 100 genes predictions. There are various intersections of predicted genes between the models.

REVIEWERS' COMMENTS:

Reviewer #2 (Remarks to the Author):

The authors have addressed my concerns and I don't have any further comments. Thank you!

Reviewer #3 (Remarks to the Author):

I thank the authors for performing additional experiments. I know that these can be quite a bit of work, but I think they helped to clarify the manuscript quite a bit. In particular, the additional text on page 7 of the manuscript and Figure S7 provide the necessary context for interpreting the machine learning results.

My main takeaway is that the the AUCs and the selected features are quite variable across cross-validation folds. This is not unexpected given the size of the dataset and the number of features. That is,

- xgboost and lasso regression have the same performance
- different features are selected in each cv fold, and model

With the additional information, the manuscript is clear enough to publish in my opinion. The informed reader is able to judge the utility of the models.

However, since the feature selected by the model are so variable, can it really be said that the feature rankings are actually useful for identifying AD associated genes?

Ultimately, I think this is a decision for the editor.